



# BIS-4D: Mapping soil properties and their uncertainties at 25 m resolution in the Netherlands

Anatol Helfenstein[1,2], Vera L. Mulder[1], Mirjam J.D. Hack-ten Broeke[2], Maarten van Doorn[3,4],
Kees Teuling[2], Dennis J.J. Walvoort[2], and Gerard B.M. Heuvelink[1,5]

[1]Soil Geography and Landscape Group, Wageningen University, PO Box 47, 6700 AA Wageningen, The Netherlands
[2]Soil, Water and Land Use Team, Wageningen Environmental Research, Droevendaalsesteeg 3, 6708 RC Wageningen, The Netherlands
[3]Nutriënten Management Instituut, Nieuwe Kanaal 7C, 6709 PA, Wageningen, the Netherlands
[4]Environmental Systems Analysis Group, Wageningen University, PO Box 47, 6700 AA, Wageningen, the Netherlands
[5]ISRIC – World Soil Information, PO Box 353, 6700 AJ, Wageningen, the Netherlands

**Correspondence:** Anatol Helfenstein (anatol.helfenstein@wur.nl)

**Abstract.** In response to the growing societal awareness of the critical role of healthy soils, there is an increasing demand for accurate and high-resolution soil information to inform national policies and support sustainable land management decisions. Despite advancements in digital soil mapping and initiatives like *GlobalSoilMap*, quantifying soil variability and its uncertainty across space, depth, and time remains a challenge. Therefore, maps of key soil properties are often still missing on a national scale, which is also the case in the Netherlands. To meet this challenge and fill this data gap, we introduce BIS-4D, a high resolution soil modelling and mapping platform for the Netherlands. BIS-4D delivers maps of soil texture (clay, silt and sand content), bulk density, pH, total nitrogen, oxalate-extractable phosphorus, cation exchange capacity and their uncertainties at 25 m resolution between 0 - 2 m depth in 3D space. Additionally, it provides maps of soil organic matter and its uncertainty in 3D space and time between 1953 - 2023 at the same resolution and depth range. The statistical model uses machine learning informed by soil observations numbering between 3815 - 855 950, depending on the soil property, and 366 environmental covariates. We assess the accuracy of mean and median predictions using design-based statistical inference of a probability sample and location-grouped 10-fold cross-validation, and prediction uncertainty using the prediction interval coverage probability.

We found that the accuracy of clay, sand and pH maps was highest, with the model efficiency coefficient (MEC) ranging between 0.6 - 0.92 depending on depth. Silt, bulk density, soil organic matter, total nitrogen and cation exchange capacity (MEC = 0.27 - 0.78), and especially oxalate-extractable phosphorus (MEC = -0.11 - 0.38), were more difficult to predict. One of the main limitations of BIS-4D is that prediction maps cannot be used to quantify the uncertainty of spatial aggregates. A step-by-step manual helps users decide whether BIS-4D is suitable for their intended purpose, an overview of all maps and their uncertainties can be found in the supplementary information (SI), openly available code and input data enhance reproducibility and future updates, and BIS-4D prediction maps can be easily downloaded at https://doi.org/10.4121/0c934ac6-2e95-4422-8360-d3a802766c71 (Helfenstein et al., 2024a). BIS-4D fills the previous data gap of a national scale



*GlobalSoilMap* product in the Netherlands and will hopefully facilitate the inclusion of soil spatial variability as a routine and integral part of decision support systems.

# 1 Introduction

Life on Earth, including that of humans, relies fundamentally on the availability and quality of air, water, and soil. These essential resources exhibit spatial variations in accordance with Tobler's first law of Geography, asserting that "Everything is related to everything else, but near things are more related than distant things" (Tobler, 1970). However, the spatial heterogeneity of soil properties stands out prominently over short distances compared to air and water. This disparity arises from the multifaceted nature of soil, comprising solid, liquid, and gaseous phases, rendering it less mobile and unable to create homogeneous mixtures akin to air or water. Moreover, soil formation is a gradual process unfolding over hundreds to millions of years, shaped by intricate interactions between the climate, organisms (including humans), topography, and parent material (Dokuchaev, 1899; Jenny, 1941). Some of these soil-forming factors themselves exhibit high heterogeneity over short distances. Consequently, achieving a comprehensive understanding of soil spatial variability demands a high sampling density, a task hindered by the inherent difficulty, time consumption, and expense associated with collecting soil samples. These challenges underscore the complexity of quantifying soil variation, highlighting the formidable task of mapping soils in 3D space and time (3D+T).

With the rising awareness of soil health among diverse stakeholders such as governmental bodies and value chains (Lehmann et al., 2020), soil scientists are increasingly dedicated to deliver high-resolution, accurate soil maps. Internationally prominent examples of policies for which spatio-temporal soil information is essential include several of the Sustainable Development Goals, such as "Zero hunger" and "Life on land" (United Nations, 2015) and, in Europe, the Green Deal, Common Agricultural Policy and Zero Pollution (Panagos et al., 2022). The importance of soil information for these policies has led to the EU Soil Strategy for 2030, the Soil Deal (European Commission, 2021) and most recently, the Proposal for a Directive on Soil Monitoring and Resilience (European Commission, 2023). For such policies to have an impact, it is essential that soil scientists deliver information required to facilitate land use decisions and management practices at multiple scales.

In the Netherlands (land area = $33\,481\,\text{km}^2$), the demand for soil information is also large. Located in the midst of Europe's largest delta, soils in the Netherlands are naturally very fertile (Edelmann, 1950; Römkens and Oenema, 2004). As one of Europe's most densely populated countries, multi-functional land use decisions made at national or regional level, need to be implemented at the field level, involving a broad range of diverse stakeholders. This spectrum of stakeholders collaborates on initiatives like the "Smart Land Use" project, which aims to sequester an additional $0.5\,\text{Mton}\,CO_2$-eq per year to Dutch mineral agricultural soils (Slier et al., 2023). Spatial information of soil properties can be used to evaluate soil health on Dutch agricultural fields using tools such as the Open Soil Index (OSI; Ros et al., 2022; Ros, 2023) and Soil Indicators for Agriculture (BLN 2.0; Ros et al., 2023) and for assessing soil functions at different scales (Schulte et al., 2015). Information on



soil texture and soil organic matter (SOM) are necessary for greenhouse gas reporting of the Land Use, Land Use Change and Forestry (LULUCF) sector for the United Nations Framework Convention on Climate Change and the Dutch LULUCF sub-
mission under the Kyoto Protocol (KP-LULUCF; Arets et al., 2020). Data of basic soil properties serve as inputs for modelling agricultural suitability (Mulder et al., 2022), crop precision agriculture (Been et al., 2023) and Soil-Water-Atmosphere-Plant interactions (SWAP; van Dam et al., 1997; Kroes et al., 2017). Furthermore, soil property maps contribute to initiatives such as the Watervision Agriculture and Nature (Hack-ten Broeke et al., 2019), Hydrological Instrumentations of the Netherlands (NHI, 2023) and Delta Program 2024 (Delta Programme, 2023).

Soil maps can also be used to identify and prioritize threats to soil health, as reviewed for the Netherlands by Römkens and Oenema (2004) and Hack-ten Broeke et al. (2009). Specific threats to soil health in the Netherlands include soil compaction (van den Akker and Hoogland, 2011; van den Akker et al., 2012), subsidence of peat due to oxidation and compaction (Brouwer et al., 2018; van Asselen et al., 2018), subsidence of young clay soils due to ripening on reclaimed land (Brouwer et al., 2018), and soil erosion (Hessel et al., 2011). Recently, Helfenstein et al. (2024c) mapped SOM in 3D+T, which identified decreases
in SOM at high resolution in 3D space. Spatial soil information is also crucial for agricultural businesses, both for optimizing fertilizer and manure applications for crop growth, but also for environmental accounting. The demand for such information is especially high in the Netherlands (Stokstad, 2019; Erisman, 2021; Aarts and Leeuwis, 2023), as it has the highest livestock density in the EU (Eurostat, 2022, p. 32) and ranks as the world's second-largest agricultural exporter (Jukema et al., 2023). An estimated 1 300 000 ha are phosphate saturated soils, where phosphate loss due to leaching exceeds ecologically tolerable
limits (Römkens and Oenema, 2004). Hence, providing spatially explicit soil information is crucial to adhere to Targets 4.2 and 4.3 of the Soil Deal for Europe, which aim to reduce fertilizer use by at least 20% and reduce nutrient losses by at least 50% by 2030 (European Commission, 2021). In summary, the pressure of using soils sustainably in the Netherlands is immense.

Between the 1950s and 2000, conventional soil maps were completed in many countries. Today, the well-established discipline of digital soil mapping (DSM) has been widely adopted to meet the demands for accurate and high-resolution soil infor-
mation for a wide range of purposes. Since DSM was first conceptualized (McBratney et al., 2003; Scull et al., 2003), maps of soil properties and soil types have been produced from local to global scales. These advances were propelled by initiatives like *GlobalSoilMap* (GSM) under the support of the International Union of Soil Sciences (Arrouays et al., 2014a, b, 2015) and the availability of openly accessible tutorials elucidating standard DSM workflows (Malone et al., 2017; Hengl and MacMillan, 2019; Brus et al., 2017; Brus, 2019, 2022).

Historically, the Netherlands was at the forefront of soil mapping. Scientific soil investigations in the Netherlands were started by Winand C.H. Staring in the mid-1800s followed by Jan van Baren and David J. Hissink in the early 1900s (Bouma and Hartemink, 2003). The first publication of the spatial distribution of soil properties in the Netherlands dates back to the 19th century (Felix, 1995). Systematic soil mapping became institutionalized with the establishment of the Dutch Soil Survey institute, or "Stichting voor Bodemkartering" (StiBoKa) in 1945 (Hartemink and Sonneveld, 2013). From 1950 to
1995, StiBoKa conducted conventional soil surveys (Buringh et al., 1962; de Bakker and Schelling, 1989; ten Cate et al., 1995) and produced regional maps (1:10 000 and 1:25 000 scale) and a national map (1:50 000 scale) of soil types (de Vries et al., 2003). After the development of DSM as a research field, various studies used (geo-)statistical methods to map qualitative and



quantitative soil properties using the data collected by StiBoKa (Brus and Heuvelink, 2007; Brus et al., 2009; Kempen et al., 2014; van den Berg et al., 2017; Helfenstein et al., 2022, 2024c). Several regions of the national soil map have since been updated (Kempen et al., 2009, 2011, 2012a; de Vries et al., 2014, 2017, 2018; Brouwer et al., 2018; Brouwer and Walvoort, 2019, 2020; Brouwer et al., 2021, 2023) and a variety of thematic maps were derived, such as a map of re-worked soils (Brouwer and van der Werff, 2012), a peat thickness map (Brouwer et al., 2018), a map of soil landscapes (van Delft and Maas, 2022, 2023) and the soil physical units map of the Netherlands (BOFEK; Heinen et al., 2022).

DSM has established itself and is routinely implemented across the world, but various challenges remain (Chen et al., 2022; Wadoux et al., 2021b). Maps of basic chemical, physical and especially biological soil properties are often missing (Chen et al., 2022; Wadoux et al., 2021b, challenge 8). Approximately 78 % of articles reviewed by Chen et al. (2022) mapped SOM, carbon content and carbon stocks. If a DSM product is available, predictions are often only made for one depth layer. Half of the studies reviewed by Chen et al. (2022) focused on soil properties at less than 30 cm depth only. However, users also require soil information at deeper depths and could benefit from models being able to predict at any desired depth in 3D, and for dynamic soil properties, in 3D+T (Chen et al., 2022; Wadoux et al., 2021b, challenge 5). In addition, there are numerous challenges relating to the accuracy of soil maps (Wadoux et al., 2021b, challenges 5 and 9). With regards to the accuracy, a major challenge is that the uncertainty of soil maps are often not quantified. A recent review showed that only 35% of studies mapping continuous soil properties estimated prediction uncertainty (Piikki et al., 2021). Without providing the uncertainty of a map, users cannot determine its fitness for use. Moreover, assessing map accuracy is not straightforward and involves many demanding pre-requites, for example the sampling design of the locations used for statistical validation. According to Piikki et al. (2021), only 13% of studies used probability sampling for map validation, which is the best approach for assessing map accuracy (Brus et al., 2011; Wadoux et al., 2021a; de Bruin et al., 2022). When using a soil map in a model or analysis, the uncertainty may be so large that it compromises the quality of the outputs of the model or analysis, posing risks of erroneous conclusions and decisions for end users (Knotters and Vroon, 2015; Knotters et al., 2015a, b; Heuvelink, 2018). The efficacy of uncertainty propagation analysis relies on quantifying input uncertainty realistically, emphasizing the consistent need to quantify uncertainty in soil maps. The above challenges also apply to the Netherlands, where there is not yet a product that meets all these requirements.

To meet these challenges and demands, we introduce a high resolution soil modelling and mapping platform for the Netherlands called BIS-4D (Fig. 1). It delivers maps of key soil properties according to GSM specifications and assesses their accuracy using prediction uncertainty and statistical validation. The platform provides maps of soil texture (clay, silt and sand content), bulk density (BD), pH, total nitrogen ($N_{tot}$), oxalate-extractable phosphorus ($P_{ox}$) and cation exchange capacity (CEC) at 25 m resolution between 0 and 2 m depth in 3D space (Table 1). Furthermore, we provide maps of SOM in 3D+T between 1953-2023 at the same resolution and depth range, since SOM has changed substantially over time. Note that for soil pH and SOM, specific updates were made compared to previous versions (Helfenstein et al., 2022, 2024c, Sect 2.7). These nine soil properties were chosen based on those prioritized by GSM (Arrouays et al., 2014a, b, 2015), end-user needs in the Netherlands and data availability. In collaboration with soil surveyors, database maintainers and experts on Dutch soils from Wageningen University and Research, we assess the strengths and limitations of the BIS-4D maps and recommend potential map applications. Finally,

model inputs, outputs (BIS-4D maps) and code, using free and open source software, are made available, easily accessible and well documented so that BIS-4D can be updated for future applications.

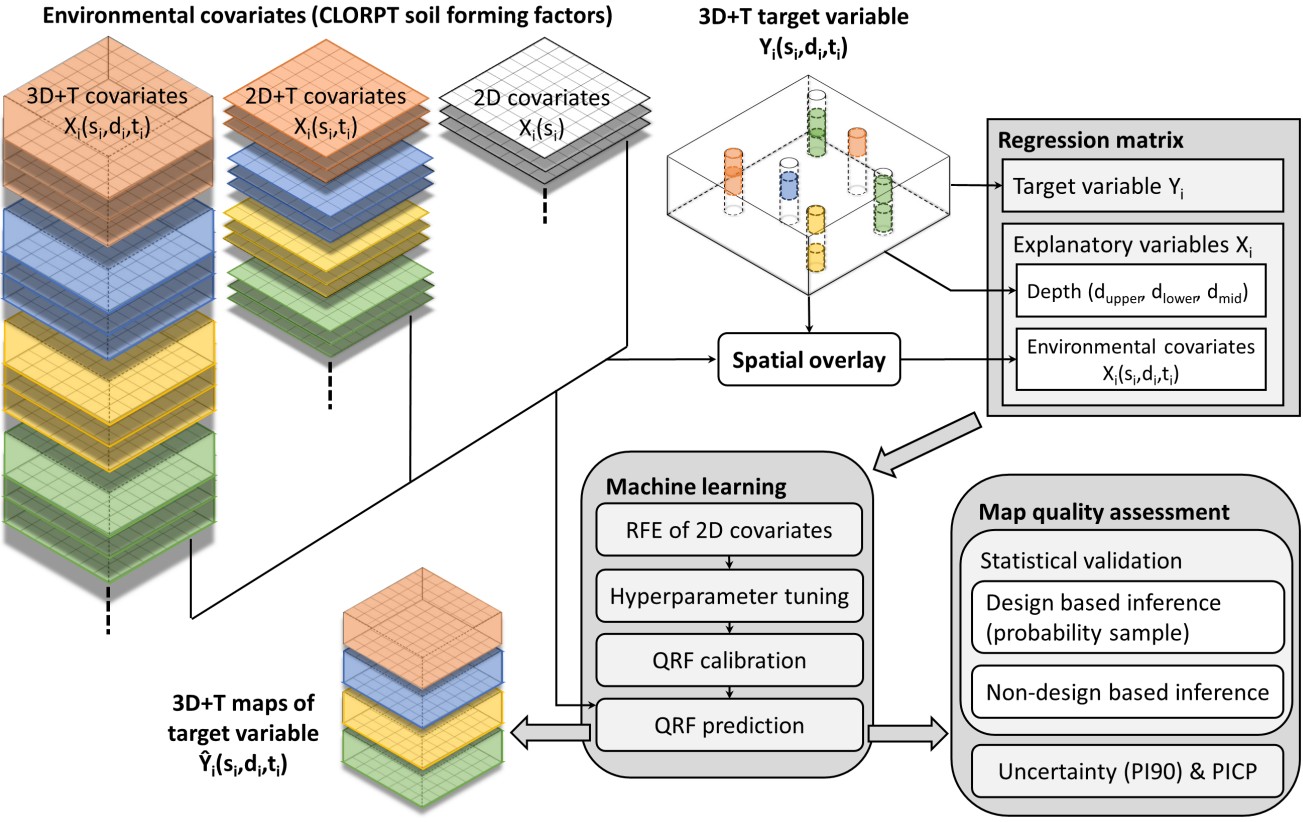

**Figure 1.** Graphical abstract of the BIS-4D soil modelling and mapping platform, where *Y* is a target soil property and *X* are covariates that vary in 2D space (*s*), depth (*d*) and, for SOM, in time (*t*). RFE = recursive feature elimination; QRF = quantile regression forest; PI90 = 90[th] prediction interval width; PICP = prediction interval coverage probability (Sect. 2.3 - 2.6).

## 125    2    Materials and Methods

We predicted soil properties $\hat{Y}$ in 3D space, and SOM in 3D+T, using well established DSM methods (Fig. 1). BIS-4D uses machine learning to model the relationship between a soil property measured at point locations as the model response *Y* (Tables 1 - 3) and environmental covariates as the explanatory variables *X* (Table 5).





**Table 1.** Abbreviations, units and description of methods used for laboratory measurements and field estimates of target soil properties. *Mineral soil is defined as the dried soil fraction (105°C) put through a 2 mm sieve after removal of SOM and $CaCO_3$.

| Soil property | Abbreviation | Unit | Description |
|---|---|---|---|
| Clay | - | % | Soil particles <2 $\mu$m as a mass percentage of the mineral soil* fraction. Measured in the laboratory using the pipette method (NEN 5753, 2020) and estimated in the field following ten Cate et al. (1995); de Bakker and Schelling (1966, 1989). |
| Silt | - | % | Soil particles 2-50 $\mu$m as a mass percentage of the mineral soil* fraction measured in the laboratory using the pipette method (NEN 5753, 2020). |
| Sand | - | % | Soil particles 50-2000 $\mu$m as a mass percentage of the mineral soil* fraction measured in the laboratory using the pipette method (NEN 5753, 2020). |
| Bulk density | BD | g/cm$^3$ | Dry bulk density of the oven-dry fine earth fraction. |
| Soil organic matter | SOM | % | Measured in the laboratory using loss on ignition at 550°C as a mass percentage of the mineral soil* fraction; or estimated in the field following ten Cate et al. (1995); de Bakker and Schelling (1966, 1989). |
| pH [KCl] | pH | - | Measured in the laboratory using pH in 1M KCl soil suspension. |
| Total N | $N_{tot}$ | mg/kg | Measured in the laboratory mainly using Jodlbauer method (Maring et al., 2009, Appendix E, p. 79). |
| Oxalate-extractable P | $P_{ox}$ | mmol/kg | Measured in the laboratory mainly using extraction with $NH_4$-oxalate at pH 3 (Maring et al., 2009, Appendix E, p. 81). |
| Cation exchange capacity | CEC | mmol(c)/kg | Measured in the laboratory mainly using extraction with silverthioureum or Ca-acetate at pH 6.5 (Maring et al., 2009, Appendix E, p. 81). |

## 2.1 Soil point data

BIS-4D uses laboratory measurements and field estimates of soil properties from point locations collected in the Dutch soil database, or "Bodemkundig informatie systeem" (BIS). Definitions and laboratory measurement and field estimation methods for the soil properties mapped using BIS-4D are described in Table 1. We only included observations between 0 and 2 m depth excluding the O horizon (humus layer).

Note that clay, silt and sand content are particle size fractions (PSF) which together constitute soil texture. Thus, soil texture
is a compositional variable: each PSF must be non-negative and together they must add up to 100% (Pawlowsky-Glahn and Buccianti, 2011; Pawlowsky-Glahn et al., 2015). In order to achieve this, soil texture can be spatially interpolated as a compositional variable using geostatistical models (Odeh et al., 2003; Lark and Bishop, 2007; Wang and Shi, 2017), e.g. compositional kriging (de Gruijter et al., 1997; Walvoort and de Gruijter, 2001), machine learning (Akpa et al., 2014; Amirian-Chakan et al., 2019; Poggio and Gimona, 2017; Poggio et al., 2021; Malone et al., 2021; Varón-Ramírez et al., 2022), and other techniques
(Buchanan et al., 2012; Román Dobarco et al., 2017). Most commonly, these studies used the additive log-ratio transforma-

tion with the Gauss-Hermite quadrature (Aitchison, 1986). When not modelled as a compositional variable, other approaches include estimating two of the three PSFs and calculating the third by subtracting the sum of the two estimates from 100% (Adhikari et al., 2013) or modelling all three PSF separately (Viscarra Rossel et al., 2015; Chagas et al., 2016; Mulder et al., 2016; Taghizadeh-mehrjardi et al., 2016; Pahlavan-Rad and Akbarimoghaddam, 2018) and post-processing the predictions
to ensure that they are all non-negative and sum to 100 %. For BIS-4D, we decided to model PSFs separately followed by post-processing (Sect 2.5) for three reasons. Firstly, we wanted to use the additional locations where only one or two PSFs were observed (Table 2). Secondly, modelling soil texture as a compositional variable does not necessarily improve model performance (Amirian-Chakan et al., 2019). Thirdly, modelling separately followed by post-processing is easy to implement.

### 2.1.1  Soil point data for model calibration

We used laboratory measurements and field estimates from the "Boring Bodemkundig pakket" (BPK) and "Profielbeschrijving" (PFB) datasets in BIS for model selection, tuning and calibration (Tables 2 & 3, Fig. 3). Observations in BPK and PFB were made by soil horizon. Laboratory measurements and field estimates were available for all depths between 0 and 2 m (Table 3). All laboratory measurements were made at PFB locations. These locations are arranged in a purposive sampling design selected in the past to create the national 1:50 000 scale soil type map (de Vries et al., 2003). For the majority of the target soil properties,
these locations covered soil variability in the Netherlands well (Fig. 2). The majority of field estimates are part of the BPK dataset and are spatially clustered in specific areas for regional soil mapping purposes or specific projects (Helfenstein et al., 2024c, Fig. 2). Most soil properties follow a skewed distribution, especially SOM, $N_{tot}$, $P_{ox}$ and CEC (Fig. 3). However, pH, sand and to a lesser extent, silt, followed bimodal distributions. The distributions of the target soil properties likely affected model predictions (Sect 3.1).

The laboratory measurements were deemed more important than field estimates because they are more accurate and locations with laboratory measurements were less spatially clustered. Nevertheless, field estimates from BPK and PFB also provide valuable information, expanding spatial coverage and, for SOM, also temporal coverage from 1953 - 2022 (Table 2). In addition, since around 2000, most observations that were added to the BIS are field estimates, a trend which is likely to continue into the future due to limited budgets for laboratory measurements. Other national mapping studies have also used field estimates
in the past (van den Berg et al., 2017). We accounted for differences in data quality between laboratory measurements and field estimates using rigorous model tuning based on optimizing model performance (Sect 2.3). Field estimates were removed if there was a laboratory measurement available from the same location and soil horizon (and year, in case of SOM). Methods for estimating clay content, BD and SOM in the field are described in ten Cate et al. (1995); de Bakker and Schelling (1966, 1989).

### 2.1.2  Soil point data for statistical validation

For clay, silt, sand and CEC, no separate dataset with laboratory measurements was available for statistical validation. Therefore, statistical validation of these four soil properties was conducted using PFB laboratory measurements and a cross-validation approach (Sect 2.6).



**Table 2.** Descriptive statistics of soil point data used for model calibration (field estimates and laboratory measurements) across all depths. Obs. = observations; Min. = minimum; Max. = maximum; Year = years during which observations were made. Minimum, median, mean and maximum values are in units of measurement of each soil property (Table 1). Soil point data used for model calibration is publicly available (Sect. 4).

| Soil property | Dataset | Method | Locations | Obs. | Min. | Median | Mean | Max. | Year |
|---|---|---|---|---|---|---|---|---|---|
| Clay | PFB | Lab | 3489 | 13 140 | 0 | 7 | 14.82 | 90.3 | 1953-2012 |
| | PFB, BPK | Field | 200 427 | 618 586 | 0 | 18 | 20.47 | 95 | 1955-2022 |
| Silt | PFB | Lab | 3376 | 12 912 | 0 | 17.8 | 24.29 | 97.5 | 1953-2002 |
| Sand | PFB | Lab | 3386 | 12 918 | 0 | 73.95 | 60.68 | 100 | 1953-2007 |
| BD | PFB | Lab | 951 | 3362 | 0.1 | 1.43 | 1.33 | 1.96 | 1957-1988 |
| | PFB, BPK | Field | 2586 | 12 509 | 0.1 | 1.5 | 1.49 | 2 | 1955-2002 |
| pH | PFB | Lab | 4216 | 15 248 | 0.9 | 4.8 | 5.2 | 9 | 1953-2010 |
| SOM | PFB | Lab | 4298 | 15 312 | 0 | 2.1 | 7 | 99.9 | 1953-2011 |
| | PFB, BPK | Field | 334 668 | 840 638 | 0 | 4 | 15.33 | 99 | 1954-2022 |
| $N_{tot}$ | PFB | Lab | 2511 | 5739 | 0 | 1300 | 3287.38 | 36 700 | 1953-2003 |
| $P_{ox}$ | PFB | Lab | 1655 | 6084 | 0 | 3.44 | 8.34 | 95.2 | 1955-2011 |
| CEC | PFB | Lab | 1332 | 3815 | 0 | 103 | 165.73 | 1541 | 1955-2010 |

**Table 3.** Number of laboratory measurements (lab) and field estimates (field) used for model calibration per standard GSM depth layer for each soil property.

| Observation type | Depth [cm] | Clay | Silt | Sand | BD | pH | SOM | $N_{tot}$ | $P_{ox}$ | CEC |
|---|---|---|---|---|---|---|---|---|---|---|
| Lab | 0-5 | 400 | 299 | 299 | 65 | 919 | 1049 | 765 | 311 | 556 |
| | 5-15 | 3844 | 3838 | 3840 | 3080 | 4524 | 5538 | 3258 | 2967 | 933 |
| | 15-30 | 1803 | 1794 | 1802 | 632 | 2519 | 2500 | 1200 | 961 | 502 |
| | 30-60 | 3731 | 3723 | 3725 | 2568 | 5392 | 5329 | 1192 | 3308 | 824 |
| | 60-100 | 4397 | 4294 | 4291 | 3630 | 5228 | 6170 | 1667 | 3972 | 728 |
| | 100-200 | 1262 | 1261 | 1258 | 1328 | 2329 | 2249 | 149 | 1566 | 272 |
| Field | 0-5 | 12 184 | - | - | 1547 | - | 18 873 | - | - | - |
| | 5-15 | 124 749 | - | - | 1372 | - | 230 710 | - | - | - |
| | 15-30 | 57 050 | - | - | 1360 | - | 117 800 | - | - | - |
| | 30-60 | 134 156 | - | - | 2242 | - | 209 918 | - | - | - |
| | 60-100 | 129 640 | - | - | 2395 | - | 138 122 | - | - | - |
| | 100-200 | 171 859 | - | - | 3593 | - | 130 836 | - | - | - |

For BD, pH, SOM, $N_{tot}$ and $P_{ox}$, laboratory measurements from either the "Landelijke Steekproef Kaarteenheden" (LSK) or "Carbon Content NL" (CCNL) dataset were available for model validation (Table 4). LSK is a separate and independent



**Figure 2.** Observation density of locations with laboratory measurements used for model calibration of all BIS-4D target soil properties. All of these locations are part of the PFB dataset.





**Figure 3.** Histograms of soil property observations used for model calibration, colored by observation type.

dataset gathered between 1993 and 2000, where locations were determined using probability sampling. The stratified simple random sample contains 94 strata defined based on soil type and groundwater class (Finke et al., 2001; Visschers et al., 2007), with the original purpose to validate the national soil type map (de Vries et al., 2003). Observations were made for each soil horizon. Statistical validation of BD, pH, SOM, $N_{tot}$ and $P_{ox}$ maps was conducted using LSK because map accuracy should preferably be estimated with design-based statistical inference using a probability sample (Brus et al., 2011). LSK data were also used to validate earlier versions of soil pH (Helfenstein et al., 2022) and SOM maps (Helfenstein et al., 2024c).

For SOM and $N_{tot}$, the CCNL dataset was used for statistical validation (Table 4). The CCNL dataset consists of all LSK locations that were still accessible in 2018. In contrast to LSK, CCNL locations were re-sampled at two fixed depth layers (0-30 cm and 30-100 cm) instead of by soil horizon. LSK and CCNL datasets were also used and their methodological sampling



**Table 4.** Descriptive statistics of separate soil point datasets used for statistical validation across all depths. Note that for statistical validation only laboratory measurements were used. Separate datasets were not available for clay, silt, sand and CEC. Obs. = observations; Min. = minimum; Max. = maximum; Year = periods during which observations were made.

| Soil property | Dataset | Locations | Obs. | Min. | Median | Mean | Max. | Year |
|---|---|---|---|---|---|---|---|---|
| BD | LSK | 1363 | 5644 | 0.17 | 1.43 | 1.29 | 1.69 | 1993-2000 |
| pH | LSK | 1363 | 5663 | 1.9 | 5.2 | 5.54 | 8.2 | 1993-2000 |
| SOM | CCNL | 1144 | 2284 | 0.5 | 3.4 | 7.51 | 78.7 | 2018 |
|  | LSK | 1185 | 4952 | 0.1 | 2.5 | 6.52 | 93.6 | 1993-2000 |
|  | $\Delta$SOM | 63 | 276 | 0 | 1.9 | 9.97 | 96.9 | 1953-1995 |
| $N_{tot}$ | CCNL | 1145 | 2286 | 0 | 1360 | 2784.85 | 24690 | 2018 |
| $P_{ox}$ | LSK | 1480 | 6220 | 0 | 3.98 | 7.05 | 96.55 | 1989-2000 |

differences were explained in van Tol-Leenders et al. (2019); van den Elsen et al. (2020); Knotters et al. (2022). Since LSK
was sampled by soil horizon, at more locations and also below 1 m depth, it is preferential to use it rather than CCNL.

For 3D+T maps of SOM, four different datasets were used for statistical validation with the specific purpose to assess SOM
maps for specific years (Helfenstein et al., 2024c): location-grouped 10-fold cross-validation of PFB data (1953-2011; lab
measurements shown in Table 2 and Fig. 3), design-based inference using LSK (1993-2000), design-based inference using
CCNL (2018) and a separate set of PFB locations that were re-sampled in 2022, used to assess changes in SOM over time
(Table 4). Design-based inference and cross-validation procedures are explained in Sect 2.6.

## 2.2 Covariates

In line with the DSM methodology (McBratney et al., 2003; Scull et al., 2003), we used 366 covariates as explanatory variables
that were representative of the soil-forming factors: climate, organisms, relief (topography), parent material (geology) and time
(Dokuchaev, 1899; Jenny, 1941). Accounting for Tobler's first law of Geography (Tobler, 1970) and spatial auto-correlation,
Easting (x-coordinate) and Northing (y-coordinate) were also included as covariates. Numerous studies have used spatial
position and geographical distances as covariates (Li et al., 2011; Behrens et al., 2018b; Hengl et al., 2018; Møller et al.,
2020; Sekulić et al., 2020). Sampling depth information, more specifically the upper and lower boundary and midpoint of each
sampled horizon, were included as covariates so that predictions could be made at any chosen depth and depth interval. See
Ma et al. (2021) for an overview of models using depth as a covariate in comparison to non-3D DSM methods. The majority
of static covariates used in BIS-4D were previously used to map soil pH (Helfenstein et al., 2022). Others, mainly derivations
of monthly mosaics from Sentinel 2 RGB and NIR bands, were added to map SOM (Helfenstein et al., 2024c). In order to map
SOM in 3D+T, we extended upon established methods by also deriving covariates variable in time (2D+T) and variable over
depth and time (3D+T), as described in detail in Helfenstein et al. (2024c). All covariates were resampled at 25 m resolution.





We created a regression matrix containing the BIS-4D target soil property observations and static covariate values by per-
forming a spatial overlay. For SOM, this was extended to a space-time overlay for 2D+T covariates and a space-depth-time
overlay for 3D+T covariates (Helfenstein et al., 2024c).

### 2.3 Model selection, tuning and calibration

For model selection as defined by Hastie et al. (2009), we removed covariates in a two-step procedure using de-correlation
followed by recursive feature elimination (RFE) as in Poggio et al. (2021). From any pair of covariates for which the Pearson
correlation coefficient was $> 0.85$ or $< -0.85$, the covariate that was more correlated with all remaining covariates was re-
moved. RFE (Guyon et al., 2002) was implemented using the `caret` package (Kuhn, 2019) and the number of covariates was
chosen with the lowest root mean squared error (RMSE; Eq. 3). From 366 covariates, this resulted in a set of 20-50 covariates
depending on the target soil property (Table 5), further used in model tuning, calibration and prediction.

For model tuning, we grew random forest (RF) models (Breiman, 2001) and optimized hyper-parameters for mean predic-
tions. We tuned the model using a location-grouped 10-fold cross-validation, meaning that all measurements from the same
soil profile location were forced to be in the same fold. Field estimates were excluded from the hold-out fold. We assessed
all combinations of the same hyper-parameters as in Sect 2.4 of Helfenstein et al. (2022) and chose the combination with the
lowest RMSE (Eq. 3, Table 6).

For soil properties where both laboratory and field estimates were available (clay, silt, sand, BD and SOM), we also tuned
whether designating a larger case weight for laboratory measurements improved model performance, in order to account for the
lower accuracy of field estimates compared to laboratory measurements. Values of two, five, ten and fifteen times the weight of
field estimates were tested for laboratory measurements (Table 6). In addition, we also tested excluding field estimates entirely.
The final set of hyper-parameters was chosen based on the lowest RMSE (Eq. 3) across the cross-validation. When the increase
in RMSE was below 0.1%, the model with fewer trees was chosen to reduce computation time during prediction. For silt and
sand, model performance was highest when using only laboratory measurements, so field estimates were excluded in model
calibration (Table 6).

For model calibration and prediction, we used RF to predict the mean and quantile regression forest (QRF) due to its ability
to predict the entire conditional distribution (Meinshausen, 2006). The final QRF used for model prediction was fitted using
all soil observations in the calibration set (Table 2), the selected covariates (Table 5) and the final set of hyper-parameters
(Table 6).

### 2.4 Variable importance

During model calibration, we assessed variable importance using the permutation method for pH, $N_{tot}$, $P_{ox}$, CEC, silt and sand,
and the impurity method for clay, BD and SOM. Permutation gives a better estimate of the variable importance than impurity
because impurity has a bias towards covariates with more distinct values, making it negatively biased towards categorical co-
variates as they have a finite number of binary splits due to their limited number of classes (Sandri and Zuccolotto, 2008, 2010).
However, the permutation measure is dependent on the out-of-bag error (Breiman, 2002). As we assigned larger weights to





**Table 5.** Covariates used during model calibration and prediction for different responses (soil properties), i.e. after covariate removal based on de-correlation and recursive feature elimination (RFE; Sect 2.3). "All" implies that a covariate was used in tuning, calibration and prediction of all soil properties. Further information can be found in the metadata files and description of the provided covariates (Sect. 4).

| Soil forming factor | Description | Source | Soil property |
|---|---|---|---|
| Soil | Peat classes starting depth and thickness | National soil map (de Vries et al., 2003) | Clay, BD, pH, $N_{tot}$, CEC |
| | Groundwater classes in agricultural areas; sub-surface material in groundwater zones | de Gruijter et al. (2004); Hoogland et al. (2014); Knotters et al. (2018) | All |
| Climate | Long-term mean, min. & max. temperature | KNMI (2020) | BD, $N_{tot}$, $P_{ox}$, CEC |
| | Long-term mean precipitation | KNMI (2020) | Clay, silt, BD, SOM, $N_{tot}$, $P_{ox}$ |
| Organism | Land use 1900, 1960, 1970, 1980 & 1986–2022 | HGN (Alterra, 2004); LGN (WENR, 2020; Hazeu et al., 2020) | Clay, silt, sand, pH, SOM, $N_{tot}$ |
| | Sentinel 2 RGB & NIR bands & spectral indices (2015-2022) as in Loiseau et al. (2019) | Roerink and Mücher (2023) | All |
| | Manure application, ammonia & total N emissions, management type | Besluit Gebruik Dierlijke Meststoffen (BGDM; RIVM, 2020); BIJ12 (2019) | Clay, silt, sand, BD, pH, $P_{ox}$, CEC |
| | Land cover & vegetation types | Bakker et al. (1989) | Clay, pH, CEC |
| | Forest vegetation types, tree species & age | de Vries and Al (1992); Clement (2001) | Clay, silt, sand, pH, $P_{ox}$, CEC |
| | Water drainage classes, areas behind dikes or not, riparian zone land cover | Maas et al. (2019) | Clay, silt, sand, pH |
| Relief | Digital elevation model (DEM) & derivatives | AHN (2023) | All |
| | Low- vs. high-elevation regions (binary) | Knotters et al. (2018) | Clay, silt, sand, pH |
| Parent material | Geological units/classes & chronostratigraphic formation period | Kombrink et al. (2012); van der Meulen et al. (2013) | Clay, silt, sand, pH, CEC |
| | Geomorphology based on geomorphological classes, genesis, form, formation time & relief | Koomen and Maas (2004); Maas et al. (2019) | Clay, silt, sand, BD, pH, SOM, $N_{tot}$, CEC |
| | Physical geographic regions & landscape types | EZK (2013) | Clay, silt, sand, BD, pH, SOM, $N_{tot}$, CEC |
| | (Paleo-) geographical maps (9000–250 B.C., 100–1850 A.D.) | Vos (2015); Vos et al. (2020) | Clay, silt, sand, BD, pH, SOM, $N_{tot}$ |
| Spatial position | Easting & Northing | - | Clay, silt, sand, BD, pH, SOM, $P_{ox}$, CEC |
| | Upper, midpoint & lower boundary of soil layer | - | All |
| Time | 2D+T dynamic covariates of land use (Helfenstein et al., 2024c) | HGN (Alterra, 2004); LGN (WENR, 2020; Hazeu et al., 2020) | SOM |
| | 2D+T & 3D+T dynamic covariates of peat classes & peat occurrence (Helfenstein et al., 2024c) | Original (1960-1995) & updated (2014-2021) national soil map (de Vries et al., 2003) | SOM |





**Table 6.** Final covariate count (post de-correlation and RFE) and optimized hyper-parameters for each modelled soil property. In instances without case weights, optimal performance was achieved excluding field estimates (silt and sand) or when the property was not estimated in the field (pH, $N_{tot}$, $P_{ox}$, and CEC).

| Soil property | Number of covariates | Number of trees | Mtry | Min. node size | Sample fraction | Split rule | Case weight |
|---|---|---|---|---|---|---|---|
| Clay | 50 | 500 | 12 | 1 | 0.8 | Variance | 5 |
| Silt | 50 | 500 | 10 | 1 | 0.8 | Variance | - |
| Sand | 50 | 500 | 10 | 1 | 0.8 | Variance | - |
| BD | 30 | 250 | 8 | 1 | 0.8 | Variance | 5 |
| pH | 50 | 500 | 12 | 1 | 0.8 | Variance | - |
| SOM | 33 | 500 | 7 | 1 | 0.8 | Variance | 10 |
| $N_{tot}$ | 20 | 500 | 4 | 1 | 0.8 | Variance | - |
| $P_{ox}$ | 30 | 500 | 6 | 1 | 0.8 | Variance | - |
| CEC | 50 | 500 | 10 | 1 | 0.63 | Variance | - |

laboratory measurements for clay, BD and SOM models, there were not enough unselected soil samples available to calculate the out-of-bag error.

### 2.5 Prediction maps

The calibrated RF and QRF and final set of covariates were used to estimate the mean, median (0.50 quantile; $q_{0.50}$), 0.05 quantile ($q_{0.05}$) and 0.95 quantile ($q_{0.95}$) at every 25 m pixel and each standard depth layer specified by GSM (0-5 cm, 5-15 cm, 15-30 cm, 30-60 cm, 60-100 cm and 100-200 cm) over the Netherlands. In addition, spatially explicit 90 % prediction interval widths (PI90) were obtained at every 25 m pixel as a measure of prediction uncertainty as follows:

$$PI90 = q_{0.95} - q_{0.05} \tag{1}$$

We post-processed the mean and median PSF prediction maps to ensure that the three PSF maps summed to 100%. The predictions of clay, silt and sand were divided by the sum of the three at that location and multiplied by 100 for every 25 m pixel.

### 2.6 Accuracy assessment

We evaluated map quality using internal (model-based) and external (model-free) accuracy assessment. At the location and 250  depth, and year in the case of SOM, of a soil property measurement, all quantiles from 0 to 1 at steps of 0.01 were predicted to obtain the PI90 (Equation 1) as well as the prediction interval coverage probability (PICP) of prediction intervals between 0.02 and 1. The PICP is the proportion of independent observations that fall into the corresponding prediction interval (Papadopoulos et al., 2001). We refer to the PICP of the PI90 as the PICP90. The PICP is an indication of how accurately QRF quantifies





uncertainty. Prediction uncertainty using PI90 is an example of a model internal accuracy assessment since it is QRF-dependent,
whereas PICP is an external accuracy metric.

Besides PICP, we used two different statistical validation methods for an external accuracy assessment: 1) design-based inference (Brus et al., 2011; Brus, 2022), using either LSK or CCNL laboratory measurements, and 2) non-design-based inference using PFB laboratory measurements (Sect 2.1.2, Table 4). We used the same approach as described in detail in Helfenstein et al. (2022) to adapt design-based inference for statistical validation of prediction maps at different depth layers. However,
design-based inference was not used to assess clay, silt, sand and CEC predictions, as it was not measured in LSK or CCNL. For non-design-based inference, we used location-grouped 10-fold cross-validation of the PFB laboratory measurements, similar as during model tuning.

To obtain commonly used accuracy metrics, both mean and median predictions were used to calculate residuals. From these residuals we estimated the mean error (ME or bias), the RMSE and the model efficiency coefficient (MEC):

$$\widehat{ME} \ = \ \frac{1}{n} \sum_{i=1}^{n} (y_i - \widehat{y_i}) \tag{2}$$

$$\widehat{RMSE} \ = \ \sqrt{\frac{1}{n} \sum_{i=1}^{n} (y_i - \widehat{y_i})^2} \tag{3}$$

$$\widehat{MEC} \ = \ 1 - \frac{\sum_{i=1}^{n} (y_i - \widehat{y_i})^2}{\sum_{i=1}^{n} (y_i - \overline{y})^2} \tag{4}$$

where $n$ is the number of validation observations, $y_i$ and $\widehat{y_i}$ are the $i$[th] observation and prediction, respectively, at a certain location, depth and year (for SOM), and $\overline{y}$ is the mean of all validation observations. Eq. 2 - 4 apply for non-design-based inference. The adapted equations for design-based inference are Eq. 5, 8 and 11 in Helfenstein et al. (2022). We computed these accuracy metrics for all observations and separated into observations pertaining to each depth layers, as the latter was necessary for design-based inference (Helfenstein et al., 2022).

In addition to rigorous quantitative accuracy assessment, we also evaluated the spatial patterns of BIS-4D prediction maps qualitatively by comparing them to existing soil maps in the Netherlands (de Vries et al., 2003; Brus et al., 2009; Schoumans and Chardon, 2015; van den Berg et al., 2017; Heinen et al., 2022; Knotters et al., 2022) and based on expert judgement. We acknowledge that qualitative evaluation was not definitive and indicative only.

## 2.7 BIS-4D updates: pH and SOM

Previous map versions of soil pH in 3D and SOM in 3D+T have recently been published using BIS-4D (Helfenstein et al., 2022, 2024c). For soil pH, this version contains several important updates. Firstly, covariates of peat classes (de Vries et al., 2003), groundwater classes in agricultural areas (Knotters et al., 2018) and Sentinel 2 RGB and NIR bands and spectral indices (Roerink and Mücher, 2023) were added, all of which were selected and thus used for model calibration and prediction of





the updated version (Table 5). We also included de-correlation and RFE to increase the signal to noise ratio and make models
more parsimonious (Sect 2.3). For 3D+T maps of SOM, we included the latest national land use map (year 2022) to derive the
dynamic 2D+T land use covariates and predict SOM for the year 2023.

## 2.8   Software and computational framework

The computational framework of BIS-4D is entirely based on open source software and was operationalized on a Ubuntu 22.04
operating system with 48 cores and 128 GB working memory (RAM). Model input data (soil point data and covariates), scripts
and model outputs (BIS-4D soil property prediction maps and their associated uncertainty maps) are openly accessible (Sect. 4).
    BIS-4D is mostly based on R (version 4.3.1; R Core Team, 2023), although GDAL (version 3.7.2; GDAL/OGR contributors,
2023) and SAGA-GIS (version 7.8.4; Conrad et al., 2015) were used during covariate preparation and processing because this
massively decreased computation time compared to using similar functions in R. Further details about resampling, masking
and processing of covariates and reclassification of categorical covariates can be found in Sect 2.7 of Helfenstein et al. (2022).
The indices necessary for the location-grouped 10-fold CV were made using the `CAST` R package (Meyer, 2023). The `caret`
package (Kuhn, 2008, 2019, 2022) was used for tuning and selection of hyper-parameters. We used the `ranger` package
(Wright and Ziegler, 2017) with the option "quantreg" to grow a QRF during calibration and without it to grow a RF during
RFE and tuning. For predictions, the option "quantiles" was used to predict quantiles while the option "response" was used
to predict the mean. A combination of the `ranger` and `terra` packages was used for predicting at all locations and depths.
We used QGIS (version 3.32.3; QGIS Development Team, 2023) and the `rasterVis` (Lamigueiro and Hijmans, 2023) and
`mapview` (Appelhans et al., 2023) R packages for exploratory and qualitative analysis and visualization of covariates and
prediction maps. The computational workflow for all BIS-4D maps took approximately 5700 CPU-hours.

## 3   Results and Discussion

BIS-4D prediction maps for every GSM depth layer at 25 m resolution can be downloaded at https://doi.org/10.4121/0c934ac6-2e95-4422-8
(Helfenstein et al., 2024a). These include predictions of the mean, 0.05, 0.50 (median) and 0.95 quantiles and the PI90 of clay,
silt, sand, BD, pH, $N_{tot}$, $P_{ox}$ and CEC. For SOM, these prediction maps are available for the years 1953, 1960, 1970, 1980,
1990, 2000, 2010, 2020 and 2023 (Sect. 4). An overview of all prediction maps together with the associated accuracy metrics
(ME, RMSE, MEC, PICP) and variable importances can be found in the supplementary information (SI), which is organized
by target soil property.

### 3.1   Accuracy assessment

### 3.1.1   Quantitative accuracy assessment

The accuracy of the produced maps varied considerably depending on the soil property (Table 7, 8 & SI). Based on 10-fold
cross-validation (Table 7), the accuracy of mean predictions over all depths for clay, sand, BD, pH and $N_{tot}$ maps was highest





(MEC > 0.70), followed by SOM and silt (MEC > 0.60). Mean predictions for $P_{ox}$ and CEC were least accurate (MEC = 0.54

and 0.49, respectively). Design-based inference separated by depth layer confirms the high accuracy of pH prediction maps (Table 8). MEC values computed for mean and median predictions using design-based inference were lower for BD (0.34 - 0.78) and $N_{tot}$ (0.27 - 0.52) than when using 10-fold cross-validation. Mean and median $P_{ox}$ maps were very inaccurate (MEC = -0.11 to 0.38) based on design-based inference. The large differences in accuracy between 10-fold cross-validation using PFB laboratory measurements and design-based inference using LSK laboratory measurements for BD and $P_{ox}$ may be

due to the clustered and limited spatial distribution of calibration data for those soil properties (Fig. 2d & h). Therefore, for BD and $P_{ox}$, metrics using 10-fold cross-validation are likely overly optimistic.

**Table 7.** Accuracy metrics of BIS-4D soil property maps using mean and median predictions, computed using 10-fold cross-validation (Sect 2.6). Units of ME and RMSE are in units of the measured soil property (Table 1).

| Soil property | ME (mean) | ME (median) | RMSE (mean) | RMSE (median) | MEC (mean) | MEC (median) | PICP90 |
|---|---|---|---|---|---|---|---|
| Clay | -0.23 | 0.42 | 8.1 | 7.7 | 0.77 | 0.78 | 0.84 |
| Silt | -0.28 | 0.59 | 12 | 13 | 0.62 | 0.57 | 0.91 |
| Sand | 0.35 | -1.2 | 17 | 17 | 0.74 | 0.74 | 0.92 |
| BD | -0.011 | -0.032 | 0.21 | 0.22 | 0.71 | 0.68 | 0.86 |
| pH | -0.010 | -0.023 | 0.71 | 0.72 | 0.73 | 0.72 | 0.93 |
| SOM | -1.0 | 0.97 | 9.5 | 9.7 | 0.64 | 0.64 | 0.88 |
| $N_{tot}$ | -37 | 390 | 2800 | 2900 | 0.72 | 0.69 | 0.91 |
| $P_{ox}$ | -0.33 | 1.5 | 7.5 | 7.7 | 0.54 | 0.52 | 0.92 |
| CEC | -3.6 | 26 | 130 | 140 | 0.49 | 0.46 | 0.92 |

The RMSE and ME were low for most soil properties (Table 7 and SI). The RMSE of sand was higher than for clay and silt, even though the MEC of sand indicates higher model performance for sand than for silt. This can be explained by the high proportion of regions in the Netherlands with very high sand content (> 75 %), i.e. the Pleistocene sandy areas shown in pink

in Fig. 4d & h. In comparison, laboratory measurements of clay and silt content were rarely > 75 % (Fig. 3).

The differences in accuracy between mean and median prediction maps varied between soil properties. Based on 10-fold cross-validation, mean predictions were less biased than median predictions for all soil properties except SOM (Table 7). For soil properties where calibration data were positively skewed (Fig. 3), i.e. all soil properties except sand, BD and pH, the bias of mean predictions was negative, whereas the bias of median predictions was positive (Table 7). However, in contrast to the

findings based on 10-fold cross-validation, design-based inference of $N_{tot}$ revealed that median predictions were less biased (between -609 and 120 mg/kg; SI) than mean predictions (between -511 and -1408 mg/kg; SI). Higher accuracy of median predicted $N_{tot}$ was also reflected in lower RMSE (Table S7) and higher MEC values (Table 8).

Mean predictions are more sensitive to extreme values and outliers than median predictions. For instance, in mineral soils, the predicted conditional distribution of SOM, $N_{tot}$, $P_{ox}$ and CEC was positively skewed and median predictions were usually

smaller than mean predictions (e.g. von Hippel, 2005, Fig. 1). In peat soils, the opposite was the case. Here, the predicted



**Table 8.** MEC for mean and median predictions of BIS-4D soil property maps, separated by depth layer and computed using either 10-fold cross-validation (CV) of PFB laboratory measurements, or design-based inference (DBI) using LSK or CCNL data (Table 4). DBI for $N_{tot}$ at 100‑200 cm depth was not possible because soil samples were not collected below 100 cm in CCNL (Sections 2.1.2 & 2.6). However, for this depth layer, CV metrics are included in the supplementary information (Table S7).

| Statistical validation method | | CV | CV | CV | DBI | DBI | DBI | DBI | DBI | CV |
|---|---|---|---|---|---|---|---|---|---|---|
| Prediction | Depth [cm] | Clay | Silt | Sand | BD | pH | SOM | $N_{tot}$ | $P_{ox}$ | CEC |
| Mean | 0-15 | 0.84 | 0.70 | 0.80 | 0.39 | 0.71 | 0.52 | 0.44 | 0.25 | 0.59 |
| | 15-30 | 0.84 | 0.68 | 0.81 | 0.78 | 0.91 | 0.53 | 0.44 | 0.17 | 0.49 |
| | 30-60 | 0.77 | 0.62 | 0.75 | 0.54 | 0.73 | 0.34 | 0.27 | -0.11 | 0.47 |
| | 60-100 | 0.69 | 0.54 | 0.67 | 0.49 | 0.74 | 0.46 | 0.27 | 0.04 | 0.38 |
| | 100-200 | 0.60 | 0.51 | 0.61 | 0.47 | 0.77 | 0.44 | - | 0.04 | 0.16 |
| Median | 0-15 | 0.84 | 0.67 | 0.79 | 0.34 | 0.71 | 0.48 | 0.52 | 0.20 | 0.56 |
| | 15-30 | 0.85 | 0.65 | 0.82 | 0.78 | 0.92 | 0.68 | 0.52 | 0.38 | 0.43 |
| | 30-60 | 0.79 | 0.58 | 0.75 | 0.54 | 0.72 | 0.27 | 0.41 | 0.05 | 0.42 |
| | 60-100 | 0.72 | 0.48 | 0.67 | 0.44 | 0.74 | 0.53 | 0.41 | 0.00 | 0.36 |
| | 100-200 | 0.63 | 0.44 | 0.61 | 0.41 | 0.76 | 0.54 | - | 0.11 | 0.26 |

conditional distribution of SOM, $N_{tot}$, $P_{ox}$ and CEC was negatively skewed and median predictions were larger than mean predictions. For these soil properties, mean predictions were thus systematically higher than median predictions in mineral soils, whereas mean predictions were systematically lower than median predictions in peat soils.

The maps of prediction uncertainty (PI90) for every GSM depth layer revealed that uncertainty was high when mean and
median predictions fell within a range with limited calibration data (SI). This meant that for most soil properties, uncertainty was high in areas where predictions were high due to the positively skewed distribution of observation data (Fig. 3). For example, the positive correlation between increasing uncertainty with increasing predictions can be clearly observed for clay and silt in Fig. 4e, f, i and j. The same positive correlation between predictions and uncertainty was observed for $N_{tot}$ over depth (Fig. 5e & f). We found a similar pattern of high uncertainty in peatlands due to high predictions in these areas for SOM,
$P_{ox}$ and CEC. However, given its bimodal distribution, the uncertainty for sand was highest in areas where predictions ranged between 25‑75 % (for example in the river areas) and uncertainty was comparatively low in marine clay areas (< 25 % sand) and high in the Pleistocene areas (> 75 % sand) (Fig. 4c, g & k). Prediction uncertainty for most soil properties increased with increasing depth (e.g. Fig. 5f), except if mean and median predictions decreased substantially over depth, as was the case for $P_{ox}$ (SI, Figs. S78-S85). Higher uncertainty at lower depths is in line with worse accuracy metrics at lower depths (Table 8;
SI) and this tendency was found in the majority of recently reviewed DSM studies (Chen et al., 2022). Finally, prediction uncertainty of most soil properties was also higher in urban areas, which can be attributed to limited soil samples and heavily disturbed soils in urban areas. With increasing population growth in an already densely populated country, this highlights the need to map urban soils (Römkens and Oenema, 2004; Vasenev et al., 2014, 2021; Kortleve et al., 2023).





**Figure 4.** Mean predicted clay [%] (a & e), silt [%] (b & f) and sand [%] content (c & g) at 60 - 100 cm depth and associated prediction uncertainty (PI90 = 90th prediction interval) and the soil physical units map of the Netherlands (BOFEK, Heinen et al., 2022, d & h) in comparison. The soil physical unit codes can be found in Heinen et al. (2022); here grouped into the main categories (1xxx = peat, 2xxx = peaty, 3xxx = sand, 4xxx = loam/clay and 5xxx = loess). The zoom-in area around Wageningen was chosen since this area contains all main soil physical categories except loess.

The PICP90 (Table 7) and the PICP (SI) indicated that prediction uncertainty was estimated relatively accurate using QRF, but small differences were found among the predicted soil properties. For clay content, the PICP90 was between 0.82 - 0.86 (Table S1) and hence less than 0.90, indicating that the uncertainty of clay predictions was underestimated. The uncertainty of BD based on PFB laboratory measurements was slightly underestimated (0.86; Table 7), but was slightly overestimated based on LSK laboratory measurements (0.88 - 0.95; Table S4). For SOM, the PICP90 varied strongly with depth (0.75 - 0.96; Table S5), but the PICP overall was very accurate for all depths combined (Fig. S53). In our study, the soil properties for which field estimates were included during calibration were the only ones for which the PI90 was sometimes underestimated.



**Figure 5.** Median predicted BD [g/cm$^3$] (a), N$_{tot}$ [mg/kg] (b), P$_{ox}$ [mmol/kg] (c), and CEC [mmol(c)/kg] (d) at 0 - 5 cm depth; and median predicted N$_{tot}$ (e) and PI90 (90$^{th}$ prediction interval) as a measure of the associated prediction uncertainty (f) along the depth transect shown in b.

Similarly, Chen et al. (2023) found that increasing the proportion of spectral estimates combined with conventional laboratory measurements decreased the PI90. Hence, if calibration data are a smoothed version of the truth, which may be the case with predictions of spectral models and field estimates, this tends to lead to underestimation of the 'true' uncertainty. The aim of sharp, i.e. narrow, conditional probability distributions by including various types of observational data is desirable



only if ensuring that the uncertainty is still reliable, e.g. by computing the PICP (Schmidinger and Heuvelink, 2023). This is important to avoid presenting overoptimistic results to end users. Besides clay, BD and SOM, prediction uncertainty for the remaining target soil properties was accurate but marginally overestimated (0.89 - 0.97) based on the independent datasets used for statistical validation (LSK and CCNL; SI). Hence, the PICP indicates that silt, sand, pH, $N_{tot}$, $P_{ox}$ and CEC maps are somewhat more accurate than suggested by the prediction uncertainty (PI90).

### 370 3.1.2 Qualitative accuracy assessment

The BIS-4D maps of the nine predicted soil properties align with the national soil map of the Netherlands (de Vries et al., 2003). This can be seen when comparing our maps (Figs. 4 & 5, SI) with the soil physical units map (BOFEK; Fig. 4d & h; Heinen et al., 2022), derived from the national soil map of the Netherlands. Further information on the comparison of BIS-4D maps to previous soil pH maps (Brus et al., 2009) can be found in Sect 4.2 of Helfenstein et al. (2022), and to previous SOM
maps (Brus et al., 2009; van den Berg et al., 2017; Knotters et al., 2022) in Helfenstein et al. (2024c). Nonetheless, visual evaluation of the maps also revealed several limitations.

The maps of soil texture or particle size fractions (clay, silt and sand) of the mineral soil component should be used with caution in peatlands, since natural peat only consists of organic matter without a mineral component. However, the low-lying fen peatlands, located mostly in the West and Northwest of the Netherlands, typically also contain some clay, silt or sometimes
even sand due to past flooding events (Edelmann, 1950; de Bakker and Schelling, 1966, 1989; Brouwer et al., 2023). Drained organic soils, particularly when under agricultural use, can also contain mineral components introduced or mixed in from mineral soil horizons from below or above the organic soil horizon. Nonetheless, 30 % clay content in a soil composed mostly of peat in absolute terms contains less clay than a mineral soil with 30 % clay content.

Visual examination of the BIS-4D maps reveals artifacts from the covariates. Although water and buildings were cropped out,
some mapping artifacts remained, such as small buildings, roads and railways. For instance, the road on top of the dike, parallel to and South of the Rhine River is clearly visible in Fig. 4e-k. This highlights the difficulty of spatial modelling approaches such as DSM that rely strongly on remote sensing products. Other artifacts were due to the combination of several Sentinel 2 images from different days in one month to obtain one monthly mosaic (Sect 2.2). Image mosaicing created artificial lines from images with more clouds, although overall, images contained very few to no clouds. Finally, there were also orthogonal artifacts most
likely due to using Northing and Easting coordinates as covariates, which can be largely removed by also including oblique axes in many additional directions (Møller et al., 2020).

### 3.2 Strengths

BIS-4D maps fill the missing data gap of spatial soil property information on a national scale in the Netherlands and bring substantial improvements to previously mapped soil properties. The main strengths of BIS-4D are: 1) the ability to provide
information of soil properties as opposed to soil types; 2) the high spatial resolution (25 m); 3) accuracy and uncertainty assessment based on best practices; 4) the benefits of machine learning combined with large amounts of data; 5) the flexibility



to predict in 3D and 3D+T; and 6) model code and data are openly available, making BIS-4D fully reproducible and easy to update.

The BIS-4D maps have several advantages compared to previous soil maps of the Netherlands. While categorical maps of soil type (de Vries et al., 2003) and derived thematic maps (Brouwer and van der Werff, 2012; Brouwer et al., 2018; van Delft and Maas, 2022, 2023; Heinen et al., 2022) are important and useful, many users require information on specific, numerical soil properties (Sect 1). We acknowledge that clay content (Brus et al., 2009), SOM (Brus et al., 2009; van den Berg et al., 2017; Knotters et al., 2022), pH (Brus et al., 2009) and soil properties related to soil texture (Heinen et al., 2022) and $P_{ox}$ (Schoumans and Chardon, 2015) have previously been mapped on a national scale in the Netherlands. However, these maps were at much coarser resolution, accuracy was either not assessed, or not assessed using design-based statistical inference, quantification and evaluation of uncertainty were missing, mapping approaches did not include machine learning, used only a few covariates, and predictions for one or several depth layers were modelled separately and only Knotters et al. (2022) assessed changes over time. The only standard GSM soil properties that we did not map are SOC, plant exploitable (effective) depth, depth to rock and coarse fragments (Arrouays et al., 2014a, b, 2015). We mapped SOM instead of SOC because, in the Netherlands, SOC was not included in routine soil analyses until recent years. However, SOC can be derived from SOM, as investigated in other studies in the Netherlands (van Tol-Leenders et al., 2019; van den Elsen et al., 2020; Teuling et al., 2021; Knotters et al., 2022). Plant exploitable (effective) depth is mostly limited by high groundwater levels in most regions of the country. Since groundwater levels have been extensively mapped in the Netherlands (de Gruijter et al., 2004; Hoogland et al., 2014; Knotters et al., 2018), mapping plant exploitable (effective) depth was not deemed necessary. Depth to rock and coarse fragments are not relevant on a national scale in the Netherlands, as the substrate materials of Dutch soils are almost exclusively either Pleistocene sand, fine-grained Quaternary sediments or peat.

Another strength of BIS-4D is that maps are at a high spatial resolution of 25 m. As covariates such as remote sensing products and national maps of land use (Hazeu et al., 2023) and digital elevation models (AHN, 2023) are nowadays available at 5 - 25 m resolution, useful information for modelling complex relationships between soil-forming factors such as land cover and topography and soil properties is provided at these scales. The increasing availability of high resolution information in soil-related domains has also increased the demand for high resolution soil maps. While high resolution products such as BIS-4D bring many advantages, it is crucial to emphasize that resolution is not an indicator of accuracy and should not be used solely to determine a map's fitness for use (de Bruin et al., 2001; Malone et al., 2013; Knotters and Walvoort, 2020; Szatmári et al., 2021).

One of the main advantages of BIS-4D is the rigorous map quality evaluation using design-based statistical inference and prediction uncertainty. Based on sampling theory (Cochran, 1977; de Gruijter et al., 2006; Gregoire and Valentine, 2007), map accuracy should be assessed with design-based statistical inference using a probability sample whenever possible, as this provides a better estimate of the "true" map accuracy compared to non-design-based approaches (Brus et al., 2011). Moreover, it also produces confidence intervals (Tables S4-S8), so that we know how close the estimate of the map accuracy is to the true map accuracy. We were able to use design-based inference for BD, pH, SOM, $N_{tot}$ and $P_{ox}$ maps due to the availability of the LSK and CCNL datasets. We are not aware of any other GSM products that used design-based inference to evaluate map





accuracy on a national scale. For soil properties for which design-based inference was not possible, i.e. for clay, silt, sand and CEC, we used location-grouped 10-fold cross-validation, as recommended in the case of non-clustered data (Wadoux et al., 2021a; de Bruin et al., 2022). In addition, BIS-4D maps provide spatially explicit estimations of prediction uncertainty (PI90),
including GSM accuracy thresholds for soil pH (Helfenstein et al., 2022), and we evaluated the accuracy of the uncertainty using PICP.

Another strength of BIS-4D, for example when compared to previous soil property maps in the Netherlands (e.g. Brus and Heuvelink, 2007; Brus et al., 2009; van den Berg et al., 2017), is that machine learning leads to more accurate predictions than other geostatistical and regression techniques. Ensemble decision tree models such as RF and QRF have repeatedly
outperformed other spatial interpolation methods (e.g. Hengl et al., 2015; Nussbaum et al., 2017; Keskin et al., 2019; Khaledian and Miller, 2020). Ensemble decision tree models are able to capture complex, non-linear relationships between the covariates and soil properties and are widely used in recent DSM studies (Vaysse and Lagacherie, 2017; Heuvelink et al., 2020; Poggio et al., 2021; Baltensweiler et al., 2021; Nussbaum et al., 2023).

BIS-4D maps for clay, silt, sand, BD, pH, $N_{tot}$, $P_{ox}$ and CEC are in 3D (between $0\text{-}2\,\mathrm{m}$ depth) and for SOM, also dynamic
(3D+T; SI and Helfenstein et al., 2024c). This fills a largely missing gap of soil information in deeper layers (Chen et al., 2022). In addition, BIS-4D can predict at any depth, as opposed to recalibrating models when mapping individual depth layers separately (Ma et al., 2021). This improves model flexibility and efficiency and a larger amount of data can be leveraged during model tuning and calibration. For example, routine agronomic soil sampling depths in the Netherlands are $0\text{-}10\,\mathrm{cm}$ for grasslands and $0\text{-}25\,\mathrm{cm}$ for croplands, thereby deviating from the GSM standard depths (Arrouays et al., 2014a, b, 2015).
Predictions and associated uncertainty for those depths can be provided using BIS-4D without recalibrating models. This is particularly useful for uncertainty, which, unlike mean and median predictions, cannot be aggregated using e.g. weighted averaging over depth layers (Sect 3.1.1). Finally, we developed innovative covariates explicit in 3D+T, presenting a novel opportunity to extend the predictive power of machine learning to 3D+T (Helfenstein et al., 2024c). This provided a new opportunity for monitoring SOM-related soil health using a method that is explicit in 3D space.
Lastly, compared to the time-consuming effort of updating conventional soil maps, DSM products such as BIS-4D can easily be extended to other soil properties in BIS and can be updated and delivered on demand (Heuvelink et al., 2010; Kempen et al., 2009, 2012b, 2015). In comparison to an earlier version for soil pH (Helfenstein et al., 2022), the number of covariates has been substantially decreased during model selection (Sect. 2.3), which benefits reproducibility and possibilities to update maps. The model code, workflow, inputs and outputs are well documented and openly available, making procedures reproducible and
easy to update (Sect 4).

## 3.3 Limitations and improvements

Uncertainty in DSM products such as BIS-4D can be linked to three overarching sources: 1) the quantity and quality of soil point data, 2) the quantity and quality of covariates, and 3) the model structure (Heuvelink, 2014, 2018). Consequently, we discuss the limitations of BIS-4D maps with regard to uncertainties linked to soil point data, covariates, and model structure
and suggest improvements to minimize these three sources of uncertainty.





### 3.3.1 Soil point data

Measurement errors and differences in measurement methods of the soil point data may have contributed to the uncertainty of BIS-4D maps. For example, Fe, Al and P extracted by oxalate extraction are considered to consist of amorphous Fe- and Al-(hydr)oxides and P bound to those oxides. However, a fraction of oxalate-extractable P in peat soils likely consist of P bound to organically complexed Fe and Al, since those are also partially extracted during the oxalate extraction (McKeague, 1967; McKeague et al., 1971; van der Zee et al., 1990; Schoumans, 2013; Schoumans and Chardon, 2015). Recent research has devised methods to quantify the uncertainty of soil laboratory measurements (van Leeuwen et al., 2021) and to incorporate these errors into machine learning algorithms (van der Westhuizen et al., 2022). Furthermore, several slightly different methods, standards and laboratory facilities were used to measure $N_{tot}$, $P_{ox}$ and CEC (Maring et al., 2009, Appendix E). This introduced uncertainty that can be minimized by standardizing laboratory measurements and procedures.

There were several limitations related to the spatial and spatio-temporal distribution of the soil point data used in BIS-4D. The calibration data of BD, $P_{ox}$ and, to a lesser extent, CEC, were spatially clustered (Fig. 2), which most likely affected mapping accuracy of those soil properties (Sect 3.1). In addition, no wet-chemical laboratory measurements were available as part of a probability sample (LSK and CCNL) for design-based statistical inference of clay, silt, sand and CEC prediction maps (Sect 2.1.2). As most of the soil point data were collected between 1950 and 2000, soil measurement age and time should be addressed also for other soil properties besides SOM (Arrouays et al., 2017). $N_{tot}$ and CEC are strongly linked to SOM and thus temporal changes may be similar to mapped SOM changes (Helfenstein et al., 2024c). BD, pH, $N_{tot}$, $P_{ox}$ and CEC likely changed due to land use and management. However, yearly variation in $P_{ox}$ is relatively small since P binds strongly to soil particles and the plant available fractions of P with short turnover times are less than 15 % of the total reversibly bound P pool (Withers et al., 2014, Fig. 3), which is what is measured with $P_{ox}$ (Lookman et al., 1995; Neyroud and Lischer, 2003). Large quantities of topsoil data are collected for agronomic surveys every four years in the Netherlands (BZK, 2022; Eurofins Agro, 2024a, b), but only a small part of these are not privacy-protected, making it challenging to incorporate in DSM approaches. Although the point data suggest good spatial coverage of most of the basic soil properties in the Netherlands, there is a major lack in repeated laboratory measurements collected using identical sampling strategies over time, as discussed in Knotters et al. (2022) and Helfenstein et al. (2024c). A consistent national soil monitoring scheme would be beneficial for modelling dynamic soil properties in 3D+T, updating static BIS-4D maps and for accuracy assessment with more recent data.

### 3.3.2 Covariates

Although BIS-4D was able to make use of a large range of high-quality, country-specific covariates (Table 5), the main variable missing in our modelling approach is more detailed land management data, which is a common challenge in DSM (Finke, 2012; Arrouays et al., 2021). Land cover and land use covariates only indirectly provided information on land management. From 2005 onwards, annual data on the specific crop type for every agricultural parcel in the Netherlands was available ("BRP Gewaspercelen"; EZK, 2019), but these were never selected among the final covariates used for model calibration (and therefore not shown in Table 5). This implies that they did not provide additional useful information for the spatial





distribution of the target soil properties at the national scale, although different drivers may be relevant at regional and local
scales (Sect 3.4). However, regardless of the scale, national crop parcel data do not capture information on management
decisions such as fertilizer inputs, liming and ploughing frequency on agricultural lands and maintaining forests and nature
areas. These management decisions are highly relevant for many of the mapped soil properties, with the exception of particle
size fractions. For example, BD is strongly dependent on the size and driving frequency of tractors on agricultural fields (Stettler
et al., 2014).

As another example, $P_{ox}$ exhibits considerable small-scale spatial variability, as discussed and made evident by the high
nugget in the semivariogram in Fig. 6 of Lookman et al. (1995). As P in the form of phosphate is bound in the soil much
stronger than N or other plant nutrients affected by the base cation saturation and CEC, there are large legacy effects due to
historic management not captured in the covariates currently used in BIS-4D. In our study, the three most important covariates
for modelling $P_{ox}$ were the covariates related to soil horizon sampling depth (Fig. S88). The relationship of $P_{ox}$ to soil depth
is supported by empirical findings of the maximum P sorption capacity decreasing with soil depth, especially in sandy soils.
Moreover, given that $P_{ox}$ map quality was poor (Table 8 and SI), the relative importance of depth suggests that the other
covariates did not explain the spatial variation of $P_{ox}$ well, likely due to missing (historic) management data. Although not
solving the problem of missing management data, one easy step to improve the accuracy of BD and $P_{ox}$ and other management-
dependent soil properties is to only map them for agricultural areas, as was done in the Netherlands for amorphous Iron- and
Aluminium-(hydr)oxides (van Doorn et al., 2024). We expect that including dynamic covariates of land management and
climate, as discussed in Helfenstein et al. (2024c), would likely also improve modelling dynamic soil properties in 3D+T.

### 3.3.3 Model structure

Despite the many advantages of using QRF for DSM (Sect 3.2), predictions may be further improved using methods such
as convolutional or recursive neural networks (deep learning; Behrens et al., 2005, 2018a; Padarian et al., 2019b; Wadoux,
2019; Wadoux et al., 2019) or transfer learning (Liu et al., 2018; Padarian et al., 2019a; Seidel et al., 2019; Helfenstein et al.,
2021; Baumann et al., 2021), defined as the process of sharing intra-domain information and rules learned by general models
to a local domain (Pan and Yang, 2010). We recommend future research to investigate the use of deep learning and transfer
learning in the Netherlands for SOM, due to the large amount of SOM data and more opportunities in accounting for differences
in observational quality (field estimates and laboratory measurements) using more complex models. However, to the best of
our knowledge, deep learning has only outperformed ensemble decision tree models when using a small number of covariates
covering only some of the soil-forming factors, from which hyper-covariates are then derived (Wadoux, 2019). Hence, deep
learning may not improve predictions in the Netherlands, where large amounts of high-quality covariates are readily available
for all soil-forming factors. In addition, quantifying model-based uncertainty using deep learning remains a challenge. Although
model-free approaches of estimating uncertainty using deep learning have been used, e.g. involving bootstrapping (Padarian
et al., 2019b; Wadoux, 2019), we are not aware of studies that have compared the accuracy of these uncertainty estimations to
QRF-based uncertainty (PI90).



One of the main limitations of the BIS-4D modelling approach is that QRF predictions cannot be used to compute the uncertainty of spatial aggregates, for example when aggregating prediction maps of different depth layers or computing average values of a soil property for a specific land use or province. This requires quantifying cross- and spatial correlation in prediction
errors, which can be accounted for by taking a multivariate or geostatistical approach (Szatmári et al., 2021; van der Westhuizen et al., 2022; Wadoux and Heuvelink, 2023).

## 3.4 Assessment scale

We recommend using BIS-4D maps on a national scale, as long as the map quality based on the provided accuracy metrics (Tables 7 & 8; SI) and prediction uncertainty (Figs. 4i-k & 5f; SI) is sufficient for the intended use. The model was developed for
the national scale for multiple land uses. Foremost, BIS-4D maps contribute to the GSM project by delivering high-resolution, 3D (and 3D+T for SOM) maps of key soil properties with quantified uncertainty according to GSM specifications for the Netherlands. The BIS-4D maps may be especially useful for initiatives that require spatially explicit soil information across all land uses and soil types of the Netherlands. This may include national contributions to United Nations and pan-European directives and policies (Panagos et al., 2022), such as the Green Deal, the Common Agricultural Policy, Zero Pollution, the EU
Soil Strategy for 2030, the Soil Deal (European Commission, 2021) and the Proposal for a Directive on Soil Monitoring and Resilience (European Commission, 2023). For example, clay, silt, sand and SOM maps can be used to improve estimates of soil-derived greenhouse gas emissions from the LULUCF sector for the Netherlands (Arets et al., 2020).

Many potential users of BIS-4D soil property prediction maps on a national scale may require information specifically for one land use and soil type. Perhaps most commonly, users may need information for agricultural soils. For example, maps of
clay, silt, sand and SOM can provide information used to estimate the carbon sequestration potential for the "Smart Land Use" project (Slier et al., 2023), which is focused specifically on mineral soils under agricultural use on a national scale. As policy makers are mostly more interested more complex soil information, such as soil health, soil functions or soil-based ecosystem services, BIS-4D maps of several soil properties separately or combined can serve as inputs for a variety of tools that assess soil health or ecosystem services. For agricultural soils, these tools include OSI (Ros et al., 2022; Ros, 2023) and BLN 2.0 (Ros
et al., 2023). Although pH, $N_{tot}$, $P_{ox}$ and CEC can be used as approximations on a national scale, pH, plant-available N and P and CEC are part of routine agronomic soil analyses. Therefore, maps of these soil properties may be more useful for forests and nature areas, where the base cation occupation of the CEC and pH should generally remain above a certain threshold to prevent Al-toxicity. For soil pH, BIS-4D predictions should be compared to predictions in Wamelink et al. (2019), who mapped soil pH in Dutch nature areas. As soil variability is linked to soil type (e.g. mineral vs. organic) and land use, we expect that
BIS-4D model predictions would improve when modelling only one land use or soil type separately. However, this was not the scale of assessment aimed for with BIS-4D.

BIS-4D maps may also be used on a regional scale, as long as the accuracy allows and no better product is available. By regional scale we mean the level of provinces, regional water authorities, which are typically composed of one or more polders or watersheds, or large municipalities. These recommendations hold true especially for clay, sand and pH, which were predicted
with higher accuracy than the other soil properties. However, regional management decisions come with social and economical

(c) Author(s) 2024. CC BY 4.0 License.



risks. The costs of poor management decisions due to the use of inaccurate or not detailed enough soil information are often several magnitudes larger than investments for conducting a more detailed regional soil survey (Knotters and Vroon, 2015; Keller et al., 2018). In agreement with Chen et al. (2022), more research is necessary in relating DSM performance indicators such as uncertainty to cost-benefit and risk assessment analysis for improving decision support. We do not recommend the use

of BIS-4D maps on a farm or field scale, as the uncertainty of predictions is most likely too high for the precision required by the farmer. Drivers of soil variation vary locally and were presumably not captured at this scale by the soil point data, covariates and model structure. As shown in Helfenstein et al. (2022), even for a soil property like soil pH, which was relatively easy to predict, less than 10 % of the map pixels were designated with one of the highest two GSM accuracy thresholds (AA and AAA). On such a local scale, we expect that the time and costs invested in a new soil survey outweigh the risks of using

inaccurate soil data (Lemercier et al., 2022).

## 3.5 BIS-4D user manual

Based on the accuracy assessment of BIS-4D maps (Sect 3.1.1), clay, sand and pH maps were most accurate. This is in agreement with Chen et al. (2022), who found that pH was the best predicted standard GSM soil property, followed by BD, PSFs (i.e. clay, silt and sand) and SOM, based on a review of 244 articles. BIS-4D map quality of silt, BD, SOM, $N_{tot}$ and

CEC were lower. For $P_{ox}$, we only recommend it as a baseline for an overview of the distribution of this soil property in the Netherlands.

Beyond these general recommendations, we have summarized the following simple chronological steps for users to help decide whether BIS-4D maps may be suitable for their intended purpose:

1. Choose one or multiple soil properties of interest.

2. Choose the depth layer(s) (0-5 cm, 5-15 cm, 15-30 cm, 30-60 cm, 60-100 cm and 100-200 cm) and, in the case of SOM, the year (1953, 1960, 1970, 1980, 1990, 2000, 2010, 2020, 2023), for which soil information is needed.

3. Consult the accuracy metrics of mean and median predictions of the soil property and depth layer of interest (Tables 7 & 8; SI), keeping in mind that the accuracy in the intended area of use may differ from the overall accuracy of the map. If the overall map quality based on these accuracy metrics is within an acceptable range for its purpose, continue to step 4.

Use accuracy metrics based on design-based statistical inference for the soil properties for which it is available, i.e. BD, pH, SOM, $N_{tot}$ and $P_{ox}$ (Table 8; SI).

4. Choose whether to use mean or median prediction maps by comparing accuracy metrics of mean and median predictions (Tables 7 & 8; SI). Consult Sect 3.1.1 for differences in mean and median predictions found using BIS-4D.

5. Download the mean and/or median prediction maps for the chosen soil property and depth layer as well as maps of the

associated uncertainty (0.05 quantile, 0.95 quantile and/or PI90) and open them using GIS software. If soil information is required for a specific area, continue to step 6.



6. Prediction uncertainty is only useful for end-users if it is reliable (Schmidinger and Heuvelink, 2023). Therefore, check whether the prediction uncertainty is reliable by consulting the PICP. If the PICP90 is close to 0.90 (Table 7) and the PICP plot close to the 1:1 line (SI), then the provided prediction uncertainty map is reliable.

7. Ideally, prediction uncertainty should also be sharp, i.e. the PI90 should be as narrow as possible (Schmidinger and Heuvelink, 2023). Decide whether the PI90 is within an acceptable range for its purpose. If possible, fitness for use can be determined by analyzing how uncertainties in BIS-4D maps propagate through the intended usage, for example for an environmental model that uses BIS-4D maps as input. Commonly used uncertainty propagation methods include the Taylor Series or Monte Carlo methods (Heuvelink, 2018).

## 4 Data and code availability

The BIS-4D soil property prediction maps at 25 m resolution can be downloaded at https://doi.org/10.4121/0c934ac6-2e95-4422-8360-d3a8 (Helfenstein et al., 2024a). Prediction maps of the mean, median, 0.05 and 0.95 quantiles and the PI90 are available for each standard depth layer specified by GSM (0-5 cm, 5-15 cm, 15-30 cm, 30-60 cm, 60-100 cm and 100-200 cm). For SOM, maps at the same resolution and for the same depth layers are available for the years 1953, 1960, 1970, 1980, 1990, 2000, 2010, 2020 and 2023.

Regarding BIS-4D model inputs, the soil point data of laboratory measurements and field estimates used during model calibration (PFB and BPK data) are publicly available at https://doi.org/10.4121/c90215b3-bdc6-4633-b721-4c4a0259d6dc (Helfenstein et al., 2024d). The georeferenced soil point data of PFB and BPK can also be viewed at https://bodemdata.nl/bodemprofielen. LSK and CCNL data used for design-based inference are not open due to privacy agreements. The pre-processed covariates that were openly available can be downloaded at 25 m resolution at https://doi.org/10.4121/6af610ed-9006-4ac5-b399- (Helfenstein et al., 2024b). This includes the majority of the covariates used for BIS-4D, with the main exception being the covariates related to the national forestry inventory, since these data are closed. A public repository of the BIS-4D code is available here: https://git.wageningenur.nl/helfe001/bis-4d. The GitLab code repository is complete with the exception of BIS database credentials and the LSK and CCNL data. All data and code is available under the CC BY 4.0 license, except for the covariates (Helfenstein et al., 2024b), which are available under the CC BY-NC-SA 4.0 license.

*Video supplement.* A short research pitch of BIS-4D is available here: https://www.youtube.com/watch?v=ENCYUnqc-wo

*Author contributions.* All authors contributed to the scientific research and writing of this paper. A.H. developed BIS-4D, prepared the data and wrote all R scripts and sections of the paper. V.L.M., G.B.M.H. and M.J.D.H.B. provided valuable inputs and suggestions during all stages of the research, including conceptualisation of the modelling framework, interpretation of results and improving the writing. M.v.D, K.T. and D.J.J.W. provided expert knowledge on various mapped soil properties within the Dutch context.



*Competing interests.* The authors declare that they have no conflict of interest.

*Acknowledgements.* We would like to thank Joop P. Okx for envisioning and realizing the need for spatially explicit soil property information in the Netherlands and his involvement in the early stages of this project. We thank Fokke de Brouwer for his expertise on soil surveying and mapping in the Netherlands. Furthermore, we thank Dorothee van Tol-Leenders and WENR colleagues for their ongoing efforts to ultimately integrate BIS-4D maps on https://bodemdata.nl. We express our utmost gratitude to all data providers. This includes decades of rigorous work by soil surveyors and scientists from the Stichting Bodemkartering (StiBoKa), the DLO Winand Staring Centre for Integrated Land, Soil and Water Research, later known as Alterra, and Wageningen Environmental Research. And lastly this includes the providers of data used to derive hundreds of different covariates for this study, made available from different sources and by various institutions. This project (WOT-04-013-010) was financed by the Dutch Ministry of Agriculture, Nature and Food Quality: https://research.wur.nl/en/projects/soil-property-mapping-wot-04-013-010.



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
