# Peer review of "BIS-4D: Mapping soil properties and their uncertainties at 25 m resolution in the Netherlands"

_Earth System Science Data, 2024_

## Author Comment (AC1)

**Response to Reviewer 1 Comments (RC1) for ESSD-2024-26**

We thank you for your comments on our manuscript and suggestions for improving our work. We have addressed all the comments. Our response (AC) to each reviewer comment (RC) are shown in bold text below.

Best regards,

Anatol Helfenstein, on behalf of all authors

General comment:

The manuscript presents a high-resolution soil mapping platform developed for the Netherlands. The authors provide descriptions of how to perform the mapping, assess the uncertainty of the maps, and discuss the strengths and limitations of the mapping platform. They describe the software and computational network used for the mapping and share:

- most of the input data used for the mapping,
- all scripts used for soil and covariate data preparation, model training and validation, and
- the derived soil maps.

A comprehensive and detailed description is provided, which can serve as a guideline and be adapted for mapping soil properties elsewhere in the world. Therefore, the presented study is likely to attract significant international interest.

The manuscript's analysis is mostly clear, except for one aspect that pertains to the qualitative accuracy assessment. It would be important to clarify whether the BOFEK and BIS-4D maps used the same "National soil map of the Netherlands" as an input layer. If they did, then using BOFEK to assess the qualitative accuracy (patterns) of BIS-4D may not be plausible. However, BOFEK could still be used to better understand patterns visible on the soil maps, as done by the authors, providing reasoning for areas with specific properties such as higher sand content or lower bulk density, etc.

Additionally, it would be informative if the authors briefly explained why they did not add residual kriging after applying the QRF prediction.

Please find a more detailed review under "Specific Comments".

**AC: Thank you for these general comments. We would like to clarify that while BOFEK used the entire national soil map of the Netherlands as a basis and starting point (https://doi.org/10.1016/j.geoderma.2022.116123), BIS-4D only used information about peat classes from the national soil map (https://doi.org/10.1038/s43247-024-01293-y, Fig. 5). Hence, we did not include any information about mineral soils from the national soil map as covariates (input layers) and therefore we maintain that comparing clay, silt and sand predictions with the BOFEK map is plausible and has an added value in addition to the quantitative, statistical accuracy assessment. However, the reviewer is correct in that there is some overlap for the peat areas, since in these areas, both the BOFEK and BIS-4D use information from the national soil map. We will clarify this better in a revised version of the manuscript by explaining that qualitative assessment of BIS-4D clay, silt and sand maps with the BOFEK map should focus on areas with mineral soils.**

**We chose not to use residual kriging for the QRF predictions because there was no spatial autocorrelation in the residuals (pure nugget effect in semivariogram). Furthermore, this would have significantly increased the computation time, especially for the soil properties with many observations (SOM and clay), and added an additional step to the methods, which are already quite complex. Spatial position was included in the model by including coordinates (Easting and Northing; Table 5) as covariates. We will add a sentence as follows at the end of section 2.5 (L247) to inform the reader: "We chose not to use kriging of the QRF prediction residuals (regression kriging) because there was no spatial autocorrelation in the residuals and to simplify the procedure."**

Specific comments:

L80-93: please add reference about mapping activity in other European countries and give a short overview about those countries which were the first ones to prepare national coverage soil map.

**AC: We appreciate the suggestion, but this would conflict with specific comment 3 of Reviewer 2 (David Rossiter), who even suggests that "The specific case of demand for soil data in NL is relevant, the paper could start there". There are many review papers of the history of soil maps in Europe (10.1016/j.geoderma.2013.05.003), digital soil mapping and the GlobalSoilMap initiative (e.g. 10.1016/j.geoderma.2021.115567), but this is beyond the scope here. We reference these papers in the manuscript in case the readers want more background knowledge.**

L96: please provide exceptions: papers that present national 3D maps including several soil chemical and physical properties.

**AC: These are provided in the review papers that we cite in this sentence from L95-96 (Chen et al., 2022 and Wadoux et al., 2021).**

Figure 1: please add meaning of CLORPT in the figure's caption.

**AC: We will add that CLORPT stands for the soil-forming factors climate, organism, relief, parent material and time in the figure caption as well as the reference (Jenny, 1941).**

L127: … measured or estimated in the field at point locations … please consider to rephrase, because in the case of clay content, BD and SOM field estimation, too are used for building the model.

**AC: We will adjust it to "measured or estimated in the field" as suggested.**

Table 1: … Measured dry bulk density … is it correct?

**AC: We prefer to not add the word "measured" as BD was both measured in the laboratory and estimated in the field.**

L133: please recheck the appropriate meaning of O horizon, it might be a layer with undecomposed or partially decomposed organic material based on FAO's terminology or add reference of the horizon definition.

**AC: We will adjust the sentence as follows to clarify: "We only included observations between 0 and 2 m depth excluding the O horizon, or the layer with dead plant**

**material, leaves, branches and other decomposing organic material on top of mineral soils."**

L153: please add that year of sampling is given in Table 2.

**AC:  We will add this.**

L157: Figure 2 shows locations of PFB, not BPK, please check it and correct text or figure caption accordingly.

**AC: This refers to Fig. 2 of Helfenstein et al., 2024c, not Fig. 2 in this manuscript. We will change the text to "(Fig. 2 in Helfenstein et al., 2024c)".**

L159: if you think that the skewness of some soil properties affected the model prediction, why didn't you transform those variables?

**AC: Please note that this sentence is not only about skewness, as we describe in the sentence before that pH, sand and silt exhibit bimodal distributions. We state that distributions of the observational data affected model predictions because this is later discussed in the discussion section. We did not transform these variables because performance did not improve and to keep the model simple.**

**We argue in the discussion that the skewness of the data influences how mean and median predictions should be interpreted, not that the skewness of data leads to a decreased model performance (what the reviewer seems to suggest). E.g. positively skewed SOM leads to general overestimation of SOM on mineral soils when using mean predictions, median predictions are likely more valuable there (L333-338). In theory, you could indeed influence this by transforming the target variable to a more normal distribution. But when using RMSE as a metric to optimize this would likely lead to an increase in model performance (for mean predictions) on mineral soils and a decrease in model performance on organic soils. Overall, this would not lead to an increased model performance. In this context, it is not only about transforming variables but also about the choice in the metric to optimize (e.g. RMSE vs MAE).**

L170: meaning of the sentence is not completely clear, does it mean that all samples were used for model training?

**AC: We will rewrite the sentences in L170-172 as follows to improve clarity:**

**"For clay, silt, sand and CEC, no separate dataset with laboratory measurements was available for statistical validation, meaning all observations were used for model calibration. Therefore, statistical validation of these four soil properties was conducted using cross-validation of PFB laboratory measurements (Sect 2.6)."**

L171: it would be clearer if the term "validation" were used exclusively for cases where independent data was available. Since independent data was not available in this instance, it would be better to use a different term than "validation".

**AC: We disagree since adding an additional term instead of validation would be both confusing and unnecessary. 10-fold cross-validation is typically considered a validation strategy in statistical modelling and in digital soil mapping. During cross-validation, model performance is evaluated in each fold on data which was not used for model calibration.**

Table 2: add the abbreviations used under column Method, i.e.: "Lab", "Field", where you explain the meaning of those.

**AC: We will make sure to include the abbreviations in the caption as follows: "Lab = laboratory measurements; Field = field estimates; …"**

L173-174: it would be informative to show sampling locations of LSK and CCNL dataset in the supplementary material.

**AC: We will refer to Fig. 1 in Helfenstein et al., 2022 (10.1016/j.geoderma.2021.115659) for the LSK locations. As written in the text, the vast majority of LSK locations were revisited during the CCNL campaign, so these locations are almost identical and there is little added value in showing both. The supplementary material is organized by target soil property and already very extensive and the main paper also already has many tables and figures in the methods section. Therefore, we decided not to include an additional figure with the CCNL locations.**

Figure 3: legend box could be reformatted, e.g.: centred, in one row and two columns.

**AC: Thanks for the suggestion, we will do so.**

L181-183: relationship between CCNL and LSK datasets is not clear please explain it a bit more.

**AC: We will make it more clear as follows:**

**"The CCNL dataset consists of laboratory measurements from re-visited LSK locations in 2018, excluding locations that were no longer accessible. In contrast to LSK, during which soil samples were taken by soil horizon, CCNL locations were re-sampled at two fixed depth layers (0-30 cm and 30-100 cm)."**

Table 4: … Obs.  = number of observations …

**AC: Thanks, we will adjust as you suggest.**

L203: please add how you defined the 25 m resolution. Does it come from the density of soil profiles available for the mapping, or resolution of a covariate which is the most important for mapping soil properties? Or is there another reason?

**AC: We will add that 25m resolution was chosen due to the resolution of the national land use maps (LGN).**

L211-213: Recursive feature elimination (RFE) is highly computationally intensive in the case of >100 possible predictors. How did you perform it? After the de-correlation method how many predictors stayed for the RFE? Based on https://git.wageningenur.nl/helfe001/bis-4d/-/blob/master/31_regression_matrix_RFE.R?ref_type=heads script RFE was performed for SOM, clay and pH. Maybe it is not the latest script. If needed, pease clarify in the text for which soil property was RFE performed, how you handled the ones for which no RFE was performed.

**AC: Indeed some further clarification could help make it more reproducible. We selected default values of covariates to retain (50, 30, 20, 15, 10), see L295 in the script you refer to. After de-correlation using the Pearson correlation coefficient, depending on the soil property, approx. 200 covariates remained. These were then subsequently reduced to 50, 30, etc. and the model with the best accuracy (RMSE) was used. This was done for all 9 target soil properties, not only SOM, clay and pH.**

**For those three, I merely noted the run time at the top of the script for benchmarking purposes. We will add the default values of covariates that we retained in the revised manuscript and mention that it is similar to the approach used in Poggio et al., 2021 (10.5194/soil-7-217-2021).**

L215-216: please shortly describe why location-grouped 10-fold cross-validation was used, what the advantage of this method is.

**AC: We will add the phrase that "Location-grouped cross-validation was chosen because observations from the same profile in both model training and validation can lead to overly optimistic model accuracy metrics".**

L219: according to Table 2 only clay content is indicated as having both field estimate and laboratory measurement. If silt and sand content did not have field estimated values, please delete them here.

**AC: The caption of Table 2 is "...soil point data used for model calibration". We tested whether including field estimates would improve model performance for all soil properties where field estimates were available (clay, silt, sand, BD and SOM). However, since performance did not improve when using silt and sand field estimates, they were not included in the final models and thus also not included in Table 2. We will keep the text as is since it is important to also inform the readership about what did not lead to model improvement.**

L224-226: here again, the sentence starting with "For silt and sand …" is not in line with Table 2 and 3. Or maybe I miss something, please make it clear in the text and tables.

**AC: Please see our comment directly above.**

L236-238: please add more details to the sentence starting with "As we assigned …". The sample fraction is 0.8 in Table 6, isn't it the value used to divide out of bag fraction? Please explain why there were not enough samples for the out-of-bag.

**AC: We will replace this last sentence (L236-238) with:**

**"When case weights are high, out-of-bag estimation is not possible because the observations with high weights are selected in the bootstrap sample of all trees, regardless of the sample fraction. Hence, we could not compute the out-of-bag error and use the permutation variable importance measure for these observations because they were never out-of-bag."**

Table 5: please add resolution of the covariates.

**AC: All covariates were resampled to 25m resolution (L203).**

L251: in the case of PICP why 0.02 is the bottom threshold value?

**AC: Thank you for catching this. Actually all quantiles between 0 and 1 were computed at steps of 0.02 (51st prediction interval – 49th prediction interval), not 0.01. Therefore, it makes sense that 0.02 was the bottom threshold value. We will correct this in the text (L250) as follows: "..., all quantiles from 0 to 1 at steps of 0.02 were predicted...".**

L258: why do you call external accuracy assessment the one that you compute on PFB dataset? PFB was used to calibrate the model, therefore would belong to internal accuracy assessment.

**AC: We disagree with the reviewer on this matter. Cross-validation is a model-independent and thus external accuracy assessment method. It can be used regardless of the model used for calibration and prediction. Please see also Table 3 in Helfenstein et al., 2022 (10.1016/j.geoderma.2021.115659).**

L317-318: please discuss the reason for having low accuracy for P-oxalate.

**AC: We explain several reasons for low accuracy of P-oxalate in paragraph 2 of Sect. 3.3.2, which makes more sense for the overall structure of the paper. This first part of the results should not go into too much detail for one soil property but rather provide an overview of the results.**

L319-321: can it be the reason that there is some difference in data quality or measurement accuracy or the way that a measurement method is performed (even if it is done with the same methodology there can be some difference in how the sample is pre-treated, etc.) that in PFIB dataset the accuracy of BD and P-oxalate is higher than in the LSK dataset?

**AC: Although this is a good point, there is no evidence that suggests that pre-treatment or measurement method was different. Instead, there is a lot of evidence to suggest that it rather has to do with the different sampling designs of the two datasets (L319-321).**

L322-325: it would be informative to compute relative error which would support further explanation.

**AC: We agree, but we did not want to introduce an additional metric simply for this single explanation (1 sentence). We think that the explanation is clear and understandable also without computing relative errors.**

Table 8: What can be the reason for higher MEC for 15-30 cm depth in the case of BD and pH? Please discuss possible reason of decreasing accuracy with depth in the case of sand, silt, and clay content around L350.

**AC: We agree that it is good to add a possible explanation on this. We will add the following in L350, after it is explained that accuracy decreased with increasing depth:**

**"Deeper soil layers are generally more difficult to predict because limited information about the subsoil can be derived from most covariates, especially remote sensing products. However, for BD and pH, the accuracy from 15-30cm depth may have been higher than from 0-15cm depth because only 245 observations were available for statistical validation in LSK from 15-30cm depth (Tables S4 & S6). Therefore, the metrics computed via design-based inference from 15-30cm depth for BD and pH are likely less representative of map quality compared to metrics of the other depth layers, where many more observations were available."**

L345-347: please add possible reasons for having higher uncertainty of the maps in river and Pleistocene areas.

**AC: As the topic sentence of this paragraph explains (L339-340), uncertainty was high when mean and median predictions fell within a range with limited calibration data. "...given its bimodal distribution, the uncertainty for sand was highest in areas where predictions ranged 345 between 25 - 75 % (for example in the river areas) and uncertainty was comparatively low in marine clay areas (< 25 % sand) and Pleistocene areas (> 75 % sand) (Fig. 4c, g & k)" as written in L345-347. Note that thanks to the comment, I have realized there was a mistake in this sentence.**

**Uncertainty was low in both marine clay areas and Pleistocene areas (and not only in marine areas), as the map in Fig. 4k clearly shows and as supported by Fig. S30 in the supplements. In the above quotation, we have corrected this. In summary, there was higher uncertainty in river areas because sand content was often between 25-75% (in this range there was little calibration data), while uncertainty was low for marine and Pleistocene areas (in those ranges there was more calibration data).**

L366: "avoid presenting overoptimistic results to end users", maybe it could be added that it is important to clarify the quality of data used for the mapping for the users of the derived maps. Just a note: uncertainty of estimated input data is higher than uncertainty of input data measured in the laboratory.

**AC: We agree with the reviewer that it is important to clarify users about the quality of the input data, but here we are arguing that accurate quantification of the prediction uncertainty is essential (L365). Therefore, we would like to keep the text as it is.**

L371-372: is there any overlap between the predictors of BIS-4D maps and information considered to prepare the soil physical units map? If both considered peat classes from the "National soil map" or and groundwater classes or land use map of HGN, LGN or geological units or geomorphology, etc. "patterns" of the maps will be similar. Please consider it and rephrase the paragraph if needed. If BOFEK is the same national soil map as indicated in Table 5 for the peat classes, then comparison of BOFEK and BIS-4D is not plausible.

**AC: Please see our first comment (AC) above. We will add the following text in the methods to explain the overlap but maintain that it is plausible to compare BIS-4D maps with BOFEK:**

**"Note that we did not compare visual patterns of the national soil map (de Vries et al., 2003) and the soil physical units map (BOFEK; Heinen et al., 2022) to BIS-4D predictions in peat areas, as covariates of peat classes were used in model calibration (Table 5 and Fig. 5 in Helfenstein et al., 2024c)."**

L378: please add proportion of peatlands areas to clarify the size of the area affected.

**AC: Good point. We will do so: "…should be used with caution in peatlands (approx. 15% of the surface area), …"**

L389: It is not clear why mosaicing created artificial lines, please describe it more or state that more analysis is needed in the future to solve it.

**AC: We have adjusted the text as follows to improve the explanation:**

**"Other artifacts were due to the combination of several Sentinel 2 images from different days in one month to obtain one monthly, cloud-free mosaic (Sect 2.2). Image mosaicing created artificial lines due to alterations in the brightness, hue and colors from images of different days."**

L503: … frequency of agricultural machinery on the fields …

**AC: We will adjust the sentence as you suggest: "For example, BD is strongly dependent on the size and driving frequency of agricultural machinery on the fields (Stettler et al., 2014).**

L510: please consider reasons coming from management – e.g.: typical depth of fertilization – and add it in the text as further possible explanation.

**AC: We will add the following sentence at the end of L510: "P from fertilization largely stays in the upper soil layers."**

L512: There are some papers on mapping soil phosphorus content at high resolution, please compare your results with those.

**AC: We will add supporting literature: "…likely due to missing (historic) management data. This is supported by other studies of mapping soil P content at high resolution (Delmas et al., 2015; Matos-Moreira et al., 2017; Kull et al., 2023).**

L550: … clay, silt, and sand content, SOM …

**AC: SOM is also a content (mass percentage). Therefore, we will correct the sentence as follows: "For example, maps of clay, silt, sand and SOM content can provide…".**

L552: … more interested in more complex …

**AC: Thanks for catching this mistake. We will correct it as follows: "are mostly interested in more complex soil information,…".**

L564, 577: … clay and sand content and pH …

**AC: Thanks for catching this mistake. We will correct it as follows: "These recommendations hold true especially for clay and sand content and pH, which…".**

L578: the BD maps of the BIS-4D were not among the most accurate ones based on the design-based inference, please rephrase the sentence.

**AC: Good point. We will adjust the sentence as follows:**

**"This is mostly in agreement with Chen et al. (2022), who found that pH was the best predicted standard GSM soil property, and PSFs (i.e. clay, silt and sand) were predicted third best, based on a review of 244 articles."**

Supplementary material:

- please add meaning of variables shown on variable importance plots, maybe as a table somewhere before Figure S11,
- please format page S37.

**AC: This information is included in table 5, where the names of the covariates are provided. Furthermore, the openly available code contains readme files for every single covariate used in BIS-4D, which allows users to easily get the covariate metadata (e.g. https://git.wur.nl/helfe001/bis-4d/-/blob/master/data/covariates/geology/geomorph2008_genese_25m_readme.txt?ref_type=heads). Lastly, this information is also part of the covariates dataset provided as an asset alongside the paper (https://doi.org/10.4121/6af610ed-9006-4ac5-b399-4795c2ac01ec).**

**However, to make this clear, we will add the phrase: "Covariate names from the y-axis can be found in code (https://git.wur.nl/helfe001/bis-4d/-**

**/tree/master/data/covariates?ref_type=heads) and covariate dataset (Helfenstein et al., 2024b).”**

**We will format page S37 in the supplements as suggested.**

---

## Author Comment (AC2)

**Response to Reviewer 2 Comments (RC2) for ESSD-2024-26**

We thank you for your comments on our manuscript and suggestions for improving our work. We have addressed all the comments. Our response (AC) to each reviewer comment (RC) are shown in bold text below.

Best regards,

Anatol Helfenstein, on behalf of all authors

General comments:

This is an outstanding contribution, not only the work itself, but also the open datasets and the comprehensive and thoughtful explanation of all the choices made in the construction of this dataset, as well as limitations and suggestions for use. I have some suggestions for improvement. But first answering the specific review questions from the journal.

-- Is there any potential of the data being useful in the future? Most certainly! Well explained in the article.

-- Are methods and materials described in sufficient detail? Yes, although anyone trying this in another country would need to be quite familiar with DSM already.

-- Are any references/citations to other data sets or articles missing or inappropriate? No

Is the article itself appropriate to support the publication of a data set? Yes, very much so.

-- Is the data set accessible via the given identifier? Yes

-- Is the data set complete? Yes

-- Are error estimates and sources of errors given (and discussed in the article)? Yes, also a very nice discussion of limitations.

-- Are the accuracy, calibration, processing, etc. state of the art? Yes, a probability sample was used for accuracy assessment, to compare with internal accuracy.

-- Are common standards used for comparison? Yes

Is the data set significant – unique, useful, and complete? Yes.

-- Consider article and data set: are there any inconsistencies within these, implausible assertions or data, or noticeable problems which would suggest the data are erroneous (or worse)? No, it is quite consistent. Problems with the source data and produced maps are well-discussed.

Is the data set itself of high quality? Very much so.

-- Is the data set usable in its current format and size? Yes

-- Are the formal metadata appropriate? No formal metadata are provided with the dataset. The user will refer to this paper to infer metadata.

-- Is the length of the article appropriate? Yes, it's long, but all is interesting.

-- Is the overall structure of the article well-structured and clear? Yes.

-- Is the language consistent and precise? Yes.

-- Are mathematical formulae, symbols, abbreviations, and units correctly defined and used? Yes.

-- Are figures and tables correct and of high quality? Yes.

Is the data set publication, as submitted, of high quality? Very much so.

**AC: Thank you very much for your positive feedback and for addressing and answering the specific review questions from the journal.**

Specific comments:

1. Neither the paper, the linked 4TU webpage description, nor the readme.txt at that site mention the coordinate reference system (EPSG:28992 Amersfoort/RD New). Yes this is given in the properties of each .tif but since this is a little-known CRS outside of NL, maybe a mention in these three places (or at least the last two) would be useful to alert the user.

**AC: Very good point. We will make sure to add the coordinate reference system in the paper and the paper assets (the three datasets and code).**

2. "soil texture" are better termed "soil particle-size separates"

**AC: We agree that soil texture can be a somewhat vague term. This is why we write "Note that clay, silt and sand content are particle size fractions (PSF) which together constitute soil texture" (L134). We henceforth always refer to PSF when speaking about clay, silt and sand together, unless we mean soil texture in a more general sense. We think that adding a third term, "soil particle-size separates", as you suggest, would create confusion and there is no real need for it. Hence, we prefer to keep "particle size fractions" (PSF) and soil texture, which are both used in numerous other papers in the field of soil science and digital soil mapping.**

3. The entire first and second paragraphs seem unnecessary in a "Data" article. Everyone knows soil data is important, this is not summary of the uses of soil data. The specific case of demand for soil data in NL is relevant, the paper could start there. The history of soil survey and databases in NL is relevant and quite interesting, it's good to see all these references collected here.

**AC: We agree that perhaps the first two paragraphs could have been shortened slightly. However, given that ESSD spans over a wide range of earth system science topics and only a small portion of these are related to soils, we think these paragraphs are key to setting the stage and help the readers understand a) how difficult it is to quantify soil information in space (and time) and b) the increasing demand for spatial soil information.**

4. L100, 101 "Wadoux et al., 2021b, challenge 5" and other references to this. Explain the "10 Challenges in Pedometrics" and where the DSM challenges come in, otherwise the reference to "challenge 5" is obscure.

**AC: We agree with the reviewer that this is now somewhat obscure. Instead of increasing the length of the introduction further, we have decided to remove the references to the specific challenge number and instead reference Wadoux et al., 2021b as a whole. That way no additional explanation is necessary.**

5. Table 1: no method is given for bulk density. Probably known-volume cores.

**AC: You are right, we only include a description for how the weight is measured, but not how density is obtained (for that the volume also needs to be known). As you suggest, it was indeed done using known-volume cores and we will add this to Table 1.**

6. There is no discussion of the point geolocation method (obviously, the older ones were not with GPS) nor the geolocation accuracy. I think the surveyors marked their locations on the 1:50k (?) topographic maps, but these were estimates, although with fairly small field sizes maybe not so uncertain. At a certain point GPS came in -- was it used? Also, when the DPOP was not so accurate in the early days of GPS?

**AC: We agree with the reviewer that it is important to include this. We will add the following text at the end of the first paragraph of Sect. 2.1 (L133):**

**"As the majority of the soil point data were collected before modern Global Positioning Systems (Table 2), soil surveyors marked the point locations on a 1:25'000 topographic map."**

**In addition, we will add text in the discussion (Sect. 3.3.1) explaining that positional uncertainty related to soil point data most likely contributed to the overall uncertainty of BIS-4D maps. On L475, we will add the following:**

**"Positional uncertainty due to marking locations on a 1:25'000 topographic map most likely also contributed to overall uncertainty of the BIS-4D maps, as investigated in other studies (Carré et al., 2007; Grimm and Behrens, 2010)."**

**Carré, F., McBratney, A.B., Minasny, B., 2007. Estimation and potential improvement of the quality of legacy soil samples for digital soil mapping. Geoderma 141, 1–14. doi: 10.1016/j.geoderma.2007.01.018.**
**Grimm, R., Behrens, T., 2010. Uncertainty analysis of sample locations within digital soil mapping approaches. Geoderma 155, 154–163. doi: 10.1016/j.geoderma.2009.05.006.**

7. L275 "In addition to rigorous quantitative accuracy assessment, we also evaluated the spatial patterns of BIS-4D prediction maps qualitatively by comparing them to existing soil maps in the Netherlands ... and based on expert judgement." This comes then in \S3.1.2, but there is no discussion of detail patterns. For example looking at BD50 around Wehl (GLD) I see some fields well-delineated but others with some in-field detail, which don't seem to follow obvious drainage lines. Center the map at (209850, 440850) to view. I am sure there are many other locations the authors could choose to comment on the detailed pattern.

**AC: We thank the reviewer for this comment and the detailed observations of the maps. As expressed in the above comment, there are countless additional locations we could have chosen to comment on the detailed patterns in the maps. As the paper is already quite lengthy, we had to restrict ourselves and chose to comment the clay, silt and sand content maps in an area with perhaps the most diverse soil geography in the Netherlands (Fig. 4). These patterns are discussed in L323-325, L343, L347, L372 and in even more detail in L385-386. Furthermore we chose to visualize and discuss detailed patterns over depth and in a novel way to visualize soil variation over depth**

**(Fig. 5). Discussing even more examples of detailed patterns from the potentially 54 produced soil property maps (9 soil properties, 6 depth layers) was beyond the scope of this article. In terms of the quality assessment method, the focus was on properly presenting and discussing the accuracy based on statistical validation techniques and prediction uncertainty, as this can be quantified.**

8. L487, 614. What is the nature of the privacy agreement? Can these points be shared under certain circumstances/agreements?

**AC: We do not know the details of the privacy agreements and whether there are options to share topsoil data collected during agronomic surveys, as we did not use these ourselves (L487). Perhaps the companies who collect such data can provide more information, which to the best of our knowledge is mostly Eurofins Agro. As for the LSK and CCNL data (L614), more information can be obtained from Wageningen Environmental Research.**

9. The produced maps have blank areas -- most are water and sealed urban areas (building footprints) but there are others, e.g., throughout the Montferland push moraine and along the Utrechtse Heuvelrug. This should be discussed under "artefacts" L384ff.

**AC: We agree with the reviewer that we should include a remark about some areas with "no data" in the BIS-4D maps that were not water or buildings. We will do so in the suggested Sect. 3.1.2 starting at L389 with the following text:**

**"In addition, a few 25m pixels are contain no data even though they were not water or buildings. This may be due to no data values in some covariates but should be explored further in an updated version."**

10. \S3.5 "BIS-4D user manual" is hardly that -- more like "tips to the user" or "guide for proper use".

**AC: We agree. We will rename Sect. 3.5 "Best practices for proper use".**

Do the GlobalSoilMap.net specifications include P? Not according to the latest version I have, from 2015.

**AC: The reviewer is correct. Oxalate-extractable P is not a standard GSM soil property, but we also do not state that it is in the manuscript. On L119-121 we state that the nine target soil properties were chosen based on GSM, end-user needs and data availability. Hence, the decision to include oxalate-extractable P was motivated by end-user needs and data availability, even though it is not a specified GSM soil property.**

Technical corrections:

1. In typography clearly differentiate - (minus) and - (from/to), e.g. in "(MEC=-0.11-0.38)" the first - is minus, the second - from/to but they look the same

**AC: Thanks for the suggestion. We will adjust it based on LaTeX best practices. There should be a space before and after minuses, no space after a negative sign, and the "en dash" (--) should be used to indicate ranges. We will correct this throughout the article.**

2. L100 "In addition, there are numerous challenges relating to the accuracy of soil maps" should start a new paragraph (new idea = new paragraph).

**AC: We agree and we will start a new paragraph and remove the "in addition".**

3. L304, 606, 615 URL not fully visible; make these into references? e.g., I imported to Zotero and exported the reference (can be done to BibTeX also). Same with the GitHub code repository.

Helfenstein, A., Mulder, V. L., Hack-ten Broeke, M. J. D., van Doorn, M., Teuling, K., Walvoort, D. J. J., & Heuvelink, G. B. M. (2024). BIS-4D: Maps of soil properties and their uncertainties at 25 m resolution in the Netherlands (Versie 2) [GeoTIFF (.tif)]. [object Object]. https://doi.org/10.4121/0C934AC6-2E95-4422-8360-D3A802766C71

Helfenstein, A., Teuling, K., Walvoort, D. J. J., Hack-ten Broeke, M. J. D., Mulder, V. L., van Doorn, M., & Heuvelink, G. B. M. (2024). Georeferenced point data of soil properties in the Netherlands (Versie 3) [Tabular (.csv); text (.txt)]. [object Object]. https://doi.org/10.4121/C90215B3-BDC6-4633-B721-4C4A0259D6DC.V3

Helfenstein, A., Mulder, V. L., Hack-ten Broeke, M. J. D., van Doorn, M., Teuling, K., Walvoort, D. J. J., & Heuvelink, G. B. M. (2024). Spatially explicit environmental variables at 25m resolution for spatial modelling in the Netherlands (Versie 2) [GeoTIFF (.tif); tabular (.csv); text (.txt)]. [object Object]. https://doi.org/10.4121/6AF610ED-9006-4AC5-B399-4795C2AC01EC

**AC: In the ESSD journal recommendations, they recommended to have the link to the data paper assets (datasets and code) in the text followed by the citation and therefore we decided to follow these guidelines. We hope that the journal publication and type-writing team will take care of it. If not, we will only include the references.**

4. L615 "that were openly available" -> "that are.."

**AC: We will correct it to "that are…".**

5. Are units with the solidus (e.g., mg/kg) acceptable? Standard scientific notation uses negative powers when needed, e.g. mg kg$^{-1}$.

**AC: You are correct (https://www.earth-system-science-data.net/submission.html#math). We will change all unit notation with denominators accordingly.**

---

## Author Comment (AC3)

**Response to Reviewer 3 Comments (RC3) for ESSD-2024-26**

We thank you for your comments on our manuscript and suggestions for improving our work. We have addressed all the comments. Our response (AC) to each reviewer comment (RC) are shown in bold text below.

Best regards,

Anatol Helfenstein, on behalf of all authors

General comments

Helfenstein et al. present a soil modelling and mapping platform for the Netherlands (BIS-4D). It "delivers maps of soil texture (clay, silt and sand content), bulk density, pH, total nitrogen, oxalate-extractable phosphorus, cation exchange capacity and their uncertainties at 25 m resolution between 0 - 2 m depth in 3D space. Additionally, it provides maps of soil organic matter and its uncertainty in 3D space and time between 1953 - 2023 at the same resolution and depth range." (see Abstract).

For the prediction of the maps, random forest (RF) and quantile regression forest (QRF) were used. Depending on the target variable 20-50 covariates were selected from a total of 366 environmental covariates. Predecessor versions of the method and the pH map were presented and published before (Helfenstein et al., 2022).

Together with the manuscript, the authors provide (i) the BIS-4D maps, (ii) the code to produce the maps, (iii) the soil point data that was used as target variables in the calibration of the models, and (iv) the majority of the covariates that were used in the modeling in raster format at 25 m resolution (excluding the non-openly available data). The code is very well documented, in a clear format, clearly structured in a series of scripts and well-arranged presented at the archive. The data sets are very well and clearly documented as well. All assets were easy to access via the given identifier and ready to apply.

The method is clearly presented and it is well conceivable that the BIS-4D platform will be used in the future to update the maps, produce maps of other soil properties or further develop the mapping method.

The manuscript is well written, good to follow and the line of thoughts and arguments is comprehensible. There are a few sentences and transitions between paragraphs that made me stumble. I addressed those and provided suggestions in the specific comments.

In general, I think it is an impressive and extensive piece of work that deserves publication.

**AC: Thank you very much for your positive feedback and taking the time to go through the manuscript and assets (data sets and code) in detail.**

Specific comments

(1) Median predictions

My understanding is that you used random forest (RF) for the mean predictions and quantile regression forest (QRF) with the 0.50 quantile for the median predictions. However, I did not find it explicitly stated. Maybe, I overread it. If not, please, include it.

**AC: Please see L227, where we wrote: "For model calibration and prediction, we used RF to predict the mean and quantile regression forest (QRF) due to its ability to predict the entire conditional distribution (Meinshausen, 2006)". The entire probability distribution also includes the median, as you noted.**

(2) Prediction in different depths

I understood it such that you used all soil observations for RF model tuning (L214-218) and calibrating the final RF and QRF models (L227-230). Then you used the calibrated RF and QRF to predict the target variables for the chosen depths (L240ff). If this is correct, can you please rephrase L214-218. As it is now, I find it confusing. Here, one suggestion:

"For model calibration, we used RF to predict the mean and quantile regression forest (QRF) to predict the entire conditional distribution (Meinshausen, 2006). The final models were fitted using all soil observations in the calibration set (Table 2), the selected covariates (Table 5) and the final set of hyper-parameters (Table 6).

**AC: We see that based on the quoted text the reviewer is referring to L227-230. We will change the text as you suggested, i.e. change "The final QRF" to the "the final models", since as you correctly state it was both RF and QRF models.**

(3) Bias of predictions

L326-332: You measured the bias with the mean error (ME). Based on that you conclude "mean predictions were less biased than median predictions for all soil properties except SOM (Table 7)." and elaborate on the systematic differences between the bias of mean and median predictions.

My interpretation of the results is different. I would say, (a) the biases measured with ME are rather small for both, and as a result, (b) also the differences in bias measured with ME are rather small. More importantly, there are (c) systematic differences in the accuracy plots from the supplement, which are the other way round. The low and high values of the mean predictions are systematically biased such that the low values of the observed variable are overestimated, the high values underestimated. The median predictions are systematically less biased in this regard. This holds for all target variables, even though to different degree.

Thus, while the mean predictions are slightly less biased than the median predictions when averaging over all values, they are clearly more biased than the median predictions for the low and high values. Maybe, it is an effect of RF emphasizing the intermediate values? For my taste, the benefit with respect to the low and high values outweighs the slight gain with respect to the mean bias. RMSE and MEC are also quite on the same level for both variants. In summary, I would rather recommend to use the median predictions.

Can you please include the raised aspects (a-c) in your discussion and adapt your recommendation in case you find it appropriate?

**AC: Thank you for this comment and in-depth analysis. We very much agree with the reviewer and think this is an important improvement of the analysis of the results. We will change the text of this paragraph as follows (L326-332):**

**"The differences in accuracy between mean and median prediction maps varied slightly between soil properties. The low and high values of the mean predictions were systematically biased such that the low values of the observed soil property were overestimated and the high values underestimated, to varying degrees for different target soil properties (Fig. S10, S21, S32, S43, S54, S65, S76, S87 & S98).**

**Thus, while the mean predictions were slightly less biased than the median predictions when averaging over all values, except for SOM (Table 7), they were more biased than the median predictions for the low and high values. For soil properties where calibration data were positively skewed (Fig. 3), i.e. all soil properties except sand, BD and pH, the bias of mean predictions was negative, whereas the bias of median predictions was positive (Table 7). In contrast to the findings based on 10-fold cross-validation, design-based inference of $N_{tot}$ revealed that median predictions were less biased (between -609 and 120 mg/kg; SI) than mean predictions (between -511 and -1408 mg/kg; SI). Higher accuracy of median predicted $N_{tot}$ was also reflected in lower RMSE (Table S7) and higher MEC values (Table 8). In summary, although it depends on the use, overall we recommend to use median predictions since low and high values were less biased and ME, RMSE and MEC values for both mean and median predictions were similar."**

(4) Merging of data and comments in the scripts

Obviously, the quality of the data is vital for your work. This includes the quality of the chemical analysis as well as the preprocessing of the data. A major aspect is the correct merging of data from different sources. This I cannot evaluate, but it seems that for yourself there are still some questions marks.... At least there are some in the R-scripts in the code-repository associated with the submitted manuscript and data (https://git.wur.nl/helfe001/bis-4d). If you clarified the issues, great! If you did not, but the questionable variable is excluded for other reasons anyways, also great! In those cases, please simply update the comments in your scripts. If there are still open questions with variables that you use, especially the target variables, I think it would be of major importance to clarify those. Having written that, I am optimistic that it is merely a matter of updating the scripts.

At this point I like to take the opportunity to acknowledge strongly the effort you spend to put all the data together, document it clearly and provide it together with the well written and documented scripts in an easily accessible way. I know how cumbersome work it is. Big thumbs up!

In particular, I stumbled in the script 11_soil_PFB_BPK_LSK_prep.R upon:

- L73+338: If there are still open questions with PFB_CHE it might be better to exclude it?

- L259ff: If the difference between values of duplicate samples are large, it might be better simply to exclude both.

- L1058: The old variable name sounds rather different ("CEC_eff = SOM_KAT"). Is this correctly assigned?

**AC: Thank you for your suggestions and positive feedback on the work we did. I agree it's a good idea to remove some of the comments in the scripts which might confuse readers or people who want to use the scripts. We will do so for the final version. Regarding the specific comments the reviewer came across, the issues with PFB_CHE have been solved so we will remove this comment. Large differences in values of duplicate samples were indeed excluded. Effective cation exchange capacity (CEC_eff) was correctly assigned, as in Dutch it stands for "kationuitwisselcapaciteit ongebufferd". Assigning soil properties and renaming them to English was done in close collaboration with the database maintainers and soil surveyors (L121).**

(5) Depth variables

Could you please elaborate why you included "the upper and lower boundary and midpoint of each sampled horizon" (L197)? What is the benefit over using just one depth variable? And if

there is one, why not just use two depth variables? If you have two of them, the third is already defined, similarly, like it is for the particle size fractions.

**AC: This was investigated in Ma et al. (2021), please see 10.1016/j.geoderma.2020.114794, Sect. 2.4.2. The effect of including boundaries in addition to midpoint was that it led to more "stepped" predictions. Furthermore, when including both upper and lower boundaries, the model can potentially also account for depth thicknesses, although it we can only speculate if RF does this. However, this was not the focus of this study, so we would prefer not to elaborate more on this and other users can easily apply our methodology with only the midpoint, for example.**

(6) Extreme values

Can you please discuss the extreme low values of pH, BD and CEC, and the extreme high values of Ntot (Table 2) being used in the calibration data?

And concerning the BD values, how was such precision (0.1) measured in the field?

**AC: Our general approach to eliminating potential outliers was that there needs to be proof that the extreme value was an outlier and can be eliminated. Extreme values that for which no such proof was found were kept in the dataset to prevent data manipulation and support objectivity of the analysis. This was one criterium investigated in the exploratory analysis scripts of BIS-4D (scripts starting with "15_soil_BIS_expl_analysis..."). Therefore, such low values might seem unlikely but they are not impossible and if no evidence was found to remove them, they were kept. For example, pH [KCl] values of 0.9 have been measured in extremely acid heathland environments with sandy soils in the Netherlands. Bulk densities (BD) of 0.1 may be possible if it is 100% peat. Extremely high total N and CEC values are possible on agricultural parcels with very high fertilizer inputs. Regarding BD, these were also measured in the laboratory by weighing the dried soil divided by the known volume core. Including the known volume core will be added to the description of Table 1, as suggested by reviewer 2 (David Rossiter).**

(7) Variable importance plots

I found it interesting that for all of your target variables, except clay, there were only one or two variables outstanding in the variable importance plots. Could you please add a few lines about those best predictors and the respective soil property like you did for Pox in L508ff. (silt + sand: meststoffen_bgdm1993_25m, fgr_25m; BD: bodem50_2021_peatcode_25m; SOM: peat_xydt_25m; pH: fgr_25m; Ntot: bodem50_2021_peatcode_25m).

**AC: We appreciate the interest of the reviewer, but the aim of this manuscript was developing a general modelling framework and achieving high prediction accuracy, rather than model inference. Therefore, we prefer not to extend the length of the article but explaining these covariates in more detail. A general description of these covariates is included in Table 5 and more details can be found in the dataset of covariates and code repository provided alongside the manuscript. For model inference, there are more refined ways from the domain of explainable machine learning, such as biplots, partial dependence plots or Shapley values. Solely based on permutation or impurity metrics as provided in this manuscript, a more detailed analysis of the covariates seems highly speculative.**

(8) Smaller number of covariates

Did you compare the performances from the current set-up with a set-up with a more rigorously limited number of covariates? For example, by decreasing the cut-off value for the correlated variables to |0.8| or |0.7|. If yes, could you please add a few lines why you chose the current set-up. If no, just take it as a comment for future work / updates of BIS-4D. It might be also interesting for comparison with other methods, like deep learning methods (see your statement in L525: "... deep learning has only outperformed ensemble decision tree models when using a small number of covariates...".).

**AC: The cutoff value of 0.8 for the Pearson correlation coefficient was based on the value used in Poggio et al., 2021 (DOI: 10.5194/soil-7-217-2021). We did not compare it with other cutoff values but would be happy if other studies would like to compare this. However, we think that the recursive feature elimination (RFE) step after de-correlation might play a larger role in determining the final covariates chosen for model calibration and prediction.**

L10+11: Please rephrase such, that it is clear that, depending on the target variable, 20-50 environmental covariates were selected for each model from the 366 available ones.

**AC: We have spent a lot of time writing and re-writing this sentence to try to make it concise and clear. We decided to keep it as it is because all 366 covariates were considered in the modelling process, even if only 20-50 are selected in the final model. One of the strength of BIS-4D is the amount of data prepared for improving the performance, so we want to emphasize this in the abstract.**

Section 1 Introduction: Generally, well written, providing reasonable storyline and context.

L33-36: The last two sentences of the first paragraph are a little bit long and convoluted. Also, if "achieving a comprehensive understanding of soil spatial variability" would require a "high sampling density", it would kind of thwart your DSM approach. Please, rewrite the two sentences. For example, something like:

"Consequently, achieving a comprehensive understanding of soil spatial variability demands a high spatial resolution. The inherent difficulty, time consumption, and expense associated with collecting soil samples is thereby posing a major challenge for the task of mapping soils in 3D space and time (3D+T)."

**AC: We will change the text to the following to make it more concise and less convoluted:**

**"Fully grasping soil spatial variability requires dense sampling, but this is hindered by the difficulty, time, and expense of collecting soil samples. These challenges underscore the complexity of quantifying soil variation, highlighting the formidable task of mapping soils in 3D space and time (3D+T)."**

L37: If you follow my suggestion above, I further suggest to start the new paragraph with "However, with the raising awareness ..."

**AC: We prefer to not start the new paragraph with "however" as it is a new idea and also not in direct contrast to the previous paragraph.**

L37: What you mean with stakeholders like "value chains"?

**AC: We agree this is not quite correct, as it reads not as if value chains were a stakeholder. We will change the sentence as follows: "With the rising awareness of soil health among diverse stakeholders and within value chains (Lehmann et al.,**

2020), soil scientists are increasingly dedicated to deliver high-resolution, accurate soil maps."

Section 2.1 Soil point data: Nice overview and comprehensible placement of your work.

L150ff: Please provide a brief english description of the BPK and PFB data sets (samples from boreholes versus pits) as it is done in the README.md at https://git.wur.nl/helfe001/bis-4d.

**AC: According to database maintainers and soil surveyors, there are not only soil samples from boreholes in BPK and not only soil samples from soil pits in the PFB. These names and abbreviations were chosen decades ago and are a little confusing and do not help the reader if an English description would be provided.**

L275-278: Well written. I like that you explicitly did the qualitative evaluation as well, and that you acknowledged its limitations.

L339-354: Well elaborated.

L350-352: And lack of predictors / variables that describe urban effects / disturbances.

**AC: Although we agree with the reviewer that there was a lack of covariates as specific proxies of urban effects and disturbances, these would be of little use if there are not observation locations from these areas because then they are also not captured in model training. Therefore, we maintain that the main reason is limited soil samples in urban areas, as currently written.**

L377-378: Good to point out that "the mineral soil component should be used with caution in peatlands..."

L377-391: The provided information is clearly written and of high practical value.

L389-391: Another option would be to exclude the coordinates completely. If your idea is correct, the artifacts should vanish. Did you ever try that? It might be interesting to try it for future updates of BIS-4D. If the artifacts would vanish and if the performance of the prediction would not decrease substantially, excluding the coordinates would be the most straightforward solution.

**AC: We are open if future modelling studies would like to investigate this further, although this was largely already done by Møller et al., 2020 ([https://doi.org/10.5194/soil-6-269-2020](https://doi.org/10.5194/soil-6-269-2020)). They were beneficial for performance, otherwise they would not have been chosen as final covariates during RFE (L208-213).**

L396: The point "4) the benefits of machine learning combined with large amounts of data" is already spelled out in the other points. Or is there some other benefit from the machine learning? If so, please name it explicitly. If not, point 4) can be deleted.

**AC: We disagree with the reviewer that point 4) is already included in the other points (1-6) in the sentence in L394-398. "The benefits of machine learning combined with large amounts of data" is not related to any of the following:**

- **the ability to provide information of soil properties as opposed to soil types**
- **the high spatial resolution (25m)**
- **accuracy and uncertainty assessment based on best practices**
- **the flexibility to predict in 3D and 3D+T**

- **model code and data are openly available, making BIS-4D fully reproducible and easy to update**

**Therefore, we will keep the sentence as it is currently written (L394-398).**

L399-416: Well put into context.

L452-454: These two sentences feel a little convoluted for me. Please rephrase. Maybe also give examples of the innovative covariates that were used. Also, "Finally" is followed by "Lastly" in the following paragraph. Maybe replace "Finally" with, for example, "In addition".

**AC: We will replace "Finally" with "In addition" as suggested by the reviewer in L452. The examples of innovative covariates that were developed for modelling soil organic matter in 3D+T are provided in the citation given in this sentence (Helfenstein et al., 2024c; DOI: 10.1038/s43247-024-01293-y).**

L463-465: The sentence doubles with the sentence before. Please, rephrase / shorten it.

**AC: We agree with the reviewer that his sentence is too much repetition with the previous and we have decided the rephrase this paragraph as follows (L462-465):**

**"Uncertainty in DSM products such as BIS-4D can be linked to three overarching sources: 1) the quantity and quality of soil point data, 2) the quantity and quality of covariates, and 3) the model structure (Heuvelink, 2014, 2018). In the following, we discuss the limitations of BIS-4D maps with regard to these three sources of uncertainty and suggest improvements."**

L488-491: Good point.

Section 3.3.2 Covariates: Well reasoned.

Section 3.4. Assessment scale: Very important, well-founded and well written section of high practical value!

L576: I think this section is of high practical value and it makes sense to include it. However, as it is now, I recommend to sharpen it. For my understanding this is not a user manual. It is a mix of guidelines how to decide whether BIS-4D is helpful for the user´s intended purpose and recommendations how to use BIS-4D. Please, change the section title according to your intention and rephrase the text of the section accordingly, as well as L18 in the abstract. This affects also the following comment. Both comments can be handled together.

**AC: Thank you for the suggestion. As this was also remarked by reviewer 2 (David Rossiter), we will change the title of Sect. 3.5 to "Best practices for proper use". We think with this change in the title of the section, the text fits well, so we will not change it. However, we will also adjust the text in the abstract (L18) to "We describe best practices to help users decide whether BIS-4D is suitable for their intended purpose, …"**

L577-581: The first paragraph of the "BIS-4D user manual" should be moved somewhere else or substantially rephrased. My guess is, that the idea was to give some general recommendations which of the maps are considered reliable enough for which purpose. As it is now, you give only one such recommendation for Pox (and not several, as the beginning of the following paragraph is suggesting).

**AC: We agree this could be improved. We think that by changing the section title much confusion is resolved. Furthermore, we will include the first sentence of the second paragraph (L582-583) in the first paragraph and remove the phrase "Beyond these general recommendations". This will result in the following:**

**"For $P_{ox}$, we only recommend the produced maps for a qualitative overview of $P_{ox}$ spatial distribution in the Netherlands. We have summarized the following simple chronological steps for users to help decide whether BIS-4D maps may be suitable for their intended purpose: …".**

L580: Please, include "qualitative". For example: "we only recommend it for a qualitative overview of its spatial distribution in the Netherlands."

**AC: We will add the word "qualitative" as suggested by the reviewer, see comment directly above.**

Technical comments and suggestions

Use the percentage sign consistently with or without leading space. I personally prefer with leading space ("X %").

**AC: According to ESSD submission guidelines (https://www.earth-system-science-data.net/submission.html#math), we agree that there should be a space and will adjust this throughout the manuscript**

L18-20: I would prefer four sentences, instead of one long one. Simply replace the "," and the last ", and" with ".".

**AC: We agree this sentence is too long and will change it to three short sentences as follows (L18-21):**

**"A step-by-step manual helps users decide whether BIS-4D is suitable for their intended purpose. An overview of all maps and their uncertainties can be found in the supplementary information (SI). Openly available code and input data enhance reproducibility and future updates. BIS-4D prediction maps can be readily downloaded at https://doi.org/10.4121/ 20 0c934ac6-2e95-4422-8360-d3a802766c71 (Helfenstein et al., 2024a)."**

L20: Delete "easily" or replace with "readily".

**AC: We will adjust this as suggested (see above).**

L30: "making it less mobile and unable to form"

**AC: We will adjust it as suggested.**

L37-44 (paragraph 2): I would delete the last sentence and change the end of the first sentence to: "(Lehmann et al., 2020), there is an increasing demand for accurate high-resolution soil maps to facilitate land use decisions and management practices at multiple scales."

**AC: We have already improved this paragraph (see comments above) and will therefore not adjust it further.**

L56: I think "crop" can be deleted.

**AC: We agree and will remove the word "crop".**

L73-93: For me, these two paragraphs feel a bit like jumping back and forth. Maybe, simply change the order of the two paragraphs. Then, it is first the story of the Netherlands and its soil mapping, which transitions to the story of DSM.

**AC: We have spent considerable effort structuring the order of these paragraphs. At one point, I presented a version to my co-authors with the structure you propose (paragraph 6 followed by paragraph 5) and compared it to the current structure (paragraph 5 followed by 6), but we decided to use the current structure. This structure is more intuitive and "expected", as we first provide a general overview of soil maps and DSM worldwide (paragraph 5), followed by soil maps and DSM in the Netherlands (paragraph 6).**

L95: "Even though DSM has established itself and is routinely implemented across the world, various challenges remain."

**AC: We think it is better to use the transition word in the second clause of the sentence rather than starting a new paragraph with a transitioning phrase ("even though").**

L101: "related"

**AC: We will change "relating" to "related".**

L101+102: "... 5 and 9), in particular that the uncertainty of soil maps is often not quantified..."

**AC: We will change it as you suggest. Thank you for the suggestion.**

L104: "However, assessing map accuracy, ..."

**AC: We think moreover fits better here than however.**

L110: "consistent" can be deleted

**AC: We will delete that word.**

L122: "assessed" (I guess)

**AC: We will change it to past tense (assessed) as suggested.**

Fig2: Why "per 3 km2"? Please add space in front of "km".

**AC: We will add the space.**

L185: "preferable"

**AC: We will change "preferential" to "preferable".**

L204: You mean "design matrix"?

**AC: We are certain that "regression matrix" is the correct term in this context, as widely used and defined in statistical modelling (and digital soil mapping).**

L281: " the here presented version"

**AC: We will change "this version" to "the here presented version".**

L298: "..., while..."

**AC: We will add the comma in front of "while".**

L548: Can be shortened, for example: "Many potential users of BIS-4D may require information specifically for one land use or soil type."

**AC: We will shorten it as suggested.**

L551-554: Convoluted sentence, please rephrase in shorter sentences. As it is now, probably an "in" is missing after "interested"?

**AC: Thank you for catching the error. Indeed, the word "in" is missing.**

L557: Maybe replace "above a certain threshold" with "high enough".

**AC: We will replace it as suggested.**

L601: "for the intended purpose"

**AC: We will change it as suggested.**

Supplements

Variable importance-plots: Consider scaling the variable importances such, that they sum up to 100 % (or 1 if you prefer that). It simplifies reading / comparing the results.

**AC: We disagree that changing the scale of the x-axis in the variable importance plots simplifies the reading and helps with comparing the results. The variable importance plots simply show the relative importance of each covariate. However, the absolute values on the x-axis by themselves are insignificant and should also not be compared between models (e.g. variable importance value of covariate XY in the clay model with variable importance value of covariate XY in the pH model).**

Figure S55: Empty page on S37.

**AC: We will adjust the formatting. Thank you for the suggestion.**

Figure S98: "CEC" in the labels of the x-axis.

**AC: The labels on the x-axis in the supplementary figure S98 are already "CEC" so we do not understand this comment of the reviewer. The hat ("^") on "CEC" implies that they are predictions (in contrast to the observations on the y-axis) and is undisputed statistical notation and also used in all other predicted vs. observed plots in the supplements.**
* * *
(1) Very good that you provide EDA plots, in particular that you produced maps to identify spatial clustering / bias. In addition, I recommend to include in future work / updates of BIS-4D:

(a) time series of the target variables, to see whether there are obvious shifts or trends, and

(b) a check whether the value distributions of the target variables differ substantially between the data sources (BPK, LSK, PFB), for example with histograms or boxplots.

**AC: Thank you for your comments and suggestions. If the reviewer is suggesting time series of target soil properties at point locations, that is not possible because there is not monitoring data with sufficient quality at point support in the Netherlands (L490-491). However, BIS-4D does include space-time mapping in 3D+T (SOM maps in data assets and Helfenstein et al., 2024c, which is now published (DOI: 10.1038/s43247-024-01293-y)). This is discussed and referenced throughout the entire manuscript. This is also included in the code. We already compared distributions of calibration (BPK and PFB) and validation (LSK and CCNL) data using histograms and boxplots. See scripts starting with "15_soil_BIS_expl_analysis_target..." and scripts 35 for pH and SOM. For example, see L183 in exploratory analysis script "15_soil_BIS_expl_analysis_target_sand.Rmd" for a histogram of PFB laboratory measurements of sand content. Furthermore, PFB and LSK or CCNL data are also directly compared using histograms in script "35_model_data_expl_analysis.Rmd" (code chunks starting at L205 and L502, respectively).**

(2) Broken links:

In README.md / master level – "Model workflow (R scripts), 1. Soil data preparation":

- 15_soil_BIS_expl_analysis_LSK_CCNL.Rmd
- 15_soil_BIS_expl_analysis_target_SOC_SOM.Rmd
- 16_soil_BIS_remove_Ohorizon_outliers.R
- 22_cov_cat_recl_gdal_par.R
- 30_regression_matrix.R
- 40_train_RF_LLO_KFCV_hyperparameter_tuning.R
- 41_train_QRF_LLO_KFCV_optimal_model.R
- 50_model_evaluation_all_depths_PFB-OOB_PFB-CV_LSK.R
- 51_model_evaluation_depth_layers_PFB-OOB_PFB-CV_LSK_LSK-SRS.R
- 60_predict_QRF_soil_maps.R
- 61_map_soil_properties.R
- target_prediction_depth_GSM.R
- out/data/covariates/DEM_derivatives
- out/maps/other/SoilGrids_v2.0/SoilGrids_phh2o_model_evaluation_LSK_SRS_d.csv
- out/maps/target/pH_KCl/GeoTIFFs
- out/maps/target/pH_KCl/pdf

In "https://git.wur.nl/helfe001/bis-4d/-/blob/master/25_cov_expl_analysis_clorpt.Rmd":

- https://git.wur.nl/helfe001/bis-4d/-/blob/master/img/landuse_ede_wageningen_hgn.jpg

**AC: Thank you for the reminder. These links will be fixed. As written in the first sentences of the README.md, "This README is currently a duplicate of the BIS-3D README, which comes along the public release of the Helfenstein et al., 2022 manuscript. As such, this README is not yet complete, as some scripts, files and directories are missing a description." This will be done for the final publication.**

(3) In section "Summary of supporting scripts, files and directories": Maybe rename the second sub bullet "covariates" listed under bullet "data" to "other"

**AC: Ah indeed we noticed that both bullets under "data" are called "covariates". We will change one of these names.**

(4) In the script https://git.wur.nl/helfe001/bis-4d/-/blob/master/20_cov_prep_gdal.R it says:

**make noise raster, which we will later use for (ad-hoc) feature elimination:**
**all covariates less important (permutation/impurity) than noise covariate in RF**
**can be removed in final model calibration**

Is this used? If not, please remove it or add a comment in the script. If it is used, please add a comment in the manuscript.

**AC: This was used in the earlier development stages of BIS-4D but now the "noise" covariate is no longer used for ad-hoc feature elimination, so we will remove this comment to avoid confusion.**